# An Anti-Forensics Video Forgery Detection Method Based on Noise Transfer Matrix Analysis

**DOI:** 10.3390/s24165341

**Published:** 2024-08-18

**Authors:** Qing Bao, Yagang Wang, Huaimiao Hua, Kexin Dong, Feifei Lee

**Affiliations:** 1Institute of Intelligent Rehabilitation Engineering, University of Shanghai for Science and Technology, Shanghai 200093, China; bao002932@163.com (Q.B.); ygwang@usst.edu.cn (Y.W.); 2Institute of Forensic Science of Shanghai Municipal Public Security Bureau, Shanghai 200083, China; huahuaimiao@163.com (H.H.); dongkexin2306@163.com (K.D.)

**Keywords:** forensic science, video authenticity, integral GOP deletion, noise transfer matrix

## Abstract

The dispute over the authenticity of video has become a hot topic in judicial practice in recent years. Despite detection methods being updated rapidly, methods for determining authenticity have limitations, especially against high-level forgery. Deleting the integral group of pictures (GOP) length in static scenes could remove key information in the video, leading to unjust sentencing. Anyone can conduct such an operation using publicly available software, thus escaping state-of-the-art detection methods. In this paper, we propose a detection method based on noise transfer matrix analysis. A pyramid structure and a weight learning module are adopted to improve the detection rate and reduce the false positive rate. In total, 80 videos were examined through delicate anti-forensic forgery operations to verify the detection performance of the proposed method and three previously reported methods against anti-forensic forgery operations. In addition, two of the latest learning-based methods were included in our experiments to evaluate the proposed method. The experimental results show that the proposed method significantly improves the detection of frame deletion points compared with traditional and learning-based methods, especially in low false positive rate (FPR) intervals, which is meaningful in forensic science.

## 1. Introduction

Digital videos are captured using cameras developed for many scenarios, such as streets, schools, subways, and shopping malls. Even a cell phone can be used as a camera by anyone passing by such areas. Every minute, 100 hours’ worth of video is uploaded to YouTube [1]. Digital videos have been widely used in many fields for their intuitive characteristics.

In forensic science, tampered videos can convey inaccurate messages, which mislead the audience. When used in court, tampered video can have severe effects. For example, an attacker could delete a sequence of frames where a person is walking through an accident scene to destroy any evidence that the person was present [2]. Furthermore, digital audio–visual materials can be easily modified for frame insertion and deletion [3]. The authenticity of digital data must be confirmed before the data can be accepted. Digital watermarks and blind detection techniques offer authentication solutions. Digital watermarks rely on hardware for inserting watermark information in the recording process, and blind detection techniques have become a hot topic in forensic science. Scholars have proposed some useful standards to apply in detection. Common forms of tampering with digital video include deleting frames and cloning or duplicating frames or parts of a frame, as these operations are easy to implement and perform well [4].

Methods for video forensics can be extended to intra-frame and inter-frame forgery detection [5,6]. Intra-frame detection focuses on identifying the forged region in the frame, while inter-frame detection aims to determine temporal domain forgery, such as frame deletion or frame insertion. As reported by Milani et al. [7], detection methods can be divided into three categories: recording equipment artifacts, geometric/physical inconsistencies, and recoding traces. With the rapid development of neural networks, multiple networks have been used to detect video forgery [8,9,10,11,12,13,14,15].

The basis of equipment artifact detection is the unique characteristics of the device. Noise is a simple aspect that illustrates equipment artifacts. Pixel intensity is corrupted by photon counting, readout, and quantization noise. Alograini et al. [16] conducted sequential and patch analyses based on noise for object removal detection in video. Fayyaz et al. [17] developed an improved surveillance video forgery detection technique to address forged frames with induced sensor pattern noise. If the edited video has frames inserted from the same device or only consecutive frames removed, methods that rely on recording equipment artifacts [11,16,18,19,20,21,22,23] become useless for verifying the video. 

Detecting geometric/physical inconsistencies is a common method for detecting multiple features that exhibit consistency throughout the entire video. Inter-frame forgery will disrupt the consistency. Li et al. [5] proposed a novel frame deletion detection method based on optical flow orientation variation. Robust principal component analysis is applied to extract moving objects from video sequences and compute their descriptors. They examined more than 300 real-world videos to evaluate the method’s performance. The results demonstrated that the true positive rate reached 90.12% with a 7.71% false alarm rate. Fadl et al. [24] proposed a fused inter-frame forgery detection method based on the histogram of oriented gradients (HOG) and motion energy image (MEI). Shehnaz et al. [25] proposed a histogram similarity pattern-detection approach by combining HOG and local binary patterns (LBPs), using a supervised support vector machine (SVM) classifier that is trained to detect video tampering based on histogram similarity metrics. This method could recognize all forms of inter-frame forgery. Most detections of geometric/physical inconsistencies [5,8,9,10,24,26,27,28,29,30,31,32] can be fooled by removing frames from static scenes, because an abnormal moving trace is the key point to detect inconsistencies. 

It should be pointed out that some reported approaches [1,33,34] can detect frame deletion in static scenes, including integer length group of pictures (GOP) deletion. This has significant meanings in forensics, as frame deletion in static scenes is a common way to tamper with videos with anti-forensics effects. A recoding trace is an inevitable feature in a forged video, as a tampered video will be re-encoded at least once after interpolation. Wang et al. [35] proposed an algorithm to detect double compression after frame removal. An unusual phenomenon occurs in the prediction error curve of the predictive frame after some frames are dropped from the video: some frames follow the deletion from one GOP sequence to another. Suppose the predictive frame moves from one to another. In that case, the prediction error of the predictive frame will periodically increase for the relationship between the predictive frame and the intra-frame destroyed by deletion manipulation. This is solid proof that multiple compressions have occurred in the video. Similar approaches with modified models on this basis have been reported in the literature [2,36,37,38,39,40,41,42,43,44]. In recent years, network models have also been introduced to improve the performance of GOP methods. Hong et al. [42] proposed a multi-layer perceptron scheme to classify the features extracted from a GOP. Such attempts contribute to making the evaluation criteria of GOP methods more objective. 

Although existing digital forensic techniques have developed rapidly in recent decades, more approaches have achieved remarkable performance on publicly available datasets [45]. Researchers have demonstrated that many people can be deceived if forgers use anti-forensic techniques [36]. In this paper, our target to identify is the anti-forensic operation for removing the integer length of a GOP in a static scene, as shown in Figure 1. The GOP structure is described in Section 2.4. This operation can alter a video by hiding key information, which can mislead courts when making sentences. In addition, it can be easily performed by anyone who understands the principles of current detection methods. The performance of a network-based approach highly depends on the training set. No anti-forensics operation training set has been verified in network-based approaches until now. Methods for detecting equipment artifacts and geometric/physical inconsistencies can be fooled under frame deletion in static scenes. Compared with inconsistency detection methods, multiple recoding detection methods could verify the authenticity of videos even when there is frame removal in static scenes. However, such methods have two primary drawbacks:Multiple recoding traces are solid proof of a loss of originality. However, multiple recoding traces may not indicate that the video has necessarily been tampered with for reasonable transmission processes.Integral GOP frame deletion in static scenes is an effective attack against most multiple recoding detection approaches based on Wang et al.’s [35] principle.

This paper proposes an anti-forensic forged video detection method based on a noise transfer matrix analysis. The primary contributions of the proposed method are as follows:We propose a novel anti-forensic detection approach to discern forged videos (integral GOP frame deletion in static scenes), using the combination of a pyramid structure and an adaptive weight adjustment module.We adopt a pooling operation and pyramid structure to extract noise features, which are available for subsequent analysis with suppressed sensitivity.A normalization operation and the combination of successive frame results reduce the influence of variable dimensions and video recording interference.Incorporating an adaptive weight adjustment module ensures the algorithm’s universality and fast learning ability with only a single video and diverse environments, thus meeting the practical requirements of forensic science.Original videos and visual examples are discussed in the paper, which are intuitive and detailed and display the characteristics of various detection methods.The receiver operating characteristic (ROC) results demonstrate significant enhancement, particularly in the low false positive rate (FPR), indicating highly improved performance in terms of the forensic principle of no punishment in doubtful cases.

The rest of the paper is organized as follows: Section 2 presents related works on the detection of forged videos. In Section 3, the materials of the experiment are introduced, as the integer length of GOP deletion was not involved in past research. Then, we formulate the correlation of noise using a noise transfer matrix analysis. Section 4 provides the experimental results and a discussion, and lastly, conclusions are given in Section 5. 

## 2. Related Works

### 2.1. Traditional Cue-Based Detection Methods

Traditional cue-based algorithms emerged first to deal with the problem. Color histograms [26] and optical flow [27,46] could find obvious mutations in videos and could lock the suspected edit location automatically. Wan et al. [29] proposed a detection technique using a human visual system (HSV)-inspired approach. It is a low-cost and efficient method that uses subjective human evaluation for video forensics. However, the HSV method could not detect deletion in static scenes, whereas frame deletion is a common operation in video editing. Feng et al. [31] developed a motion-adaptive method to handle videos with variable motion strength. In forensics, the frame deletion point (FDP) is the key point, while the relocated I-frame (RI) has similar mean statistics. A new fluctuation feature based on frame motion residuals was developed to identify the FDP, as the specific distribution of FDP and RI are quite different. In addition, the intra-prediction elimination procedure further enhances the fluctuation feature, improving adaptability with various motion levels. The results demonstrated that the true positive rate of the algorithm could reach 90% with a false alarm rate of 0.3%. Although this method achieved excellent performance [31], its application in forensics is quite limited. The main contribution of this paper is that a motion-adaptive method is proposed, but it could not be applicable to integer length GOP frame deletion in static scenes. The method can detect FDP in motion shots rather than static scenes. 

Li et al. [5] proposed an effective descriptor, called flow orientation variation, to capture the frame deletion trace. They proved that the motion intensity along the motion trajectory in a video is an example of the wide-sense stationary process [47]. Li et al. assumed that the motion intensity variation between optical flows in a continuous frame was a first-order Markov process. Then, they created a pseudo-flow orientation variation to approximate the flow orientation variation. The flow orientation variation was determined by unbalanced intensity in the horizontal and vertical directions. Frame deletion could break the consistency of flow orientation variation, resulting in abnormal traces in videos. The positive rate reached 90.12%, while the false alarm rate was 7.71% based on 324 real-world video tests. 

Kingra et al. [32] proposed motion and brightness gradients to identify videos. Through objective analysis of prediction residual and optical flow gradients, they could locate disturbances in the temporal correlation between successive frames caused by tampering. Yao et al. [48] proposed a novel anti-forensics approach for detecting frame deletion based on interpolation analysis. They assumed that existing frame deletion detection methods could accurately locate FDPs in videos. As a result, the proposed frame interpolation at FDPs is a simple but reasonable anti-forensic strategy. The spatial and frequency domain features of interpolated and natural frame residuals are quite different. Especially, the number of interpolated frames is large. This is an ingenious approach to handling anti-forensics operations such as interpolation. In fact, the FDP in the video could be accurately detected and located, which raises doubt in forensic science. Fadl et al. [24] proposed a combined cue-based approach to detect inter-frame forgery. They used HOG and MEI to detect all forms of inter-frame forgery, including frame insertion, frame deletion, frame duplication, and frame shuffling. All of the approaches mentioned above are consecutiveness checks. Abnormal motion vectors and changed backgrounds (including linear interpolation in successive frames) are key points in detecting forgery. Delicate frame removal in static scenes is a useful way to evade detection.

### 2.2. Equipment Trace Detection Methods

Blind source camera identification is another way to determine authenticity. Bayram et al. introduced color filter array forms to classify different types of recording equipment [21]. Since common cameras are equipped with one sensor, the form of the color filter array (CFA) should be consistent in the whole image. The presence of different forms means the video has lost its originality, while the same form may not indicate originality. In addition, such an approach requires a low degree of image compression. Mandelli et al. [19] also tackled the problem with a sensor-based method. Photo response non-uniformity is a noise fingerprint characteristic of the sensor; an algorithm using this characteristic could identify the camera source. As a sensor-based method, it could not detect forged video recorded by the same device. Zhang et al. [30] established an approach to detect video forgery with ghost shadow artifacts introduced by video inpainting. The manipulation of erasing moving objects in the video would result in unusual packet loss impairments in the video, which could be evaluated by mean squared error and slice-boundary mismatch [49,50]. As the examples reported in the literature show, the approach shows little advantage over experienced inspectors who can directly observe visible artifacts. Many device identification methods involve the identification of species, not individuals. In addition, such methods cannot be applied to frame deletion detection.

The correlation of noise also plays a critical role in detecting authenticity. Hsu et al. [18] proposed a method to find forged areas in video by using noise correlation based on block-level temporal noise correlation. The noise distribution reveals the shape of the radiometric response function [23]. Kobayashi et al. [20] developed a method involving inconsistencies in noise level function (NLF) to detect forged areas in static video, which showed excellent performance. A charge-coupled device (CCD) camera converts photons into electrons, and the variance and mean of observed pixel values can be determined by some parameters. This relationship was formulated as the NLF method proposed by Liu et al. [22]. Forged videos edited by different devices may show different noise characteristics in forged regions. Similarly, Aloraini et al. [16] proposed a novel approach to detect object removal forgery in videos. They performed sequential analysis by modeling videos as stochastic processes. The value of the pixel is drawn from a normal distribution. The variance remains constant, while the mean is dependent on the scene. The combination of sequential and patch analysis is applied to detect the suspected motion trajectory of moving objects in the video. The results showed that the method achieved excellent performance with low computational complexity. 

Fayyaz et al. [17] developed an improved forgery detection technique for surveillance video based on sensor pattern noise (SPN). The SPN technique is commonly used to detect video forgery based on inaccessibility. The SPN technique requires the source of the video, which is quite hard for attackers to access. However, a problem arises when the attacker also has access to the device or knows the estimation technique used by the forensic investigator. In the method reported by Fayyaz et al. [17], a combined strategy is adopted to recognize the addition of averaged SPN forgery. In addition to the correlation of noise residue with SPN, the correlation with noise residue from the previous frame is computed to detect inconsistency in frames. The experimental results showed an average detection rate of 96% with a 4% false positive rate. 

In summary, the approaches mentioned above were designed to detect the removal of objects from video rather than frames removal. Li et al. [51] proposed a noise-based approach to detect inter-frame forgery as frame deletion in static scenes. A noise transfer matrix is introduced to check the continuity of noise features. It takes a lot of time to examine a long video, since the computation of features is at the pixel level. Moreover, the paper is more of a case report due to the lack of test samples.

### 2.3. Network-Based Detection Methods

Convolution neural networks (CNNs) and deep learning have been applied to forensic science in recent years [8,9]. Chen et al. [52] applied CNNs for median filtering in forensics. Median filtering, which preserves the edge information of an image by nonlinearity, has been utilized as an anti-forensics method. For example, the filter could destroy linear correlations between adjacent pixels to hide the trace of re-sampling [53]. This method can determine cut-and-paste forgery in low-resolution images by detecting the median filtering regions. 

Although median filtering is common in image forgery, frame deletion can be used to tamper with videos without median filtering. Pawar et al. [54] expanded image tampering detection to video based on the CNN model. To improve detection accuracy, the CNN model was used to learn and classify camera-related features extracted from images. Long et al. [8] proposed a C3D-based convolutional neural network for dropped frame detection. The C3D-based network is biased toward videos with camera motion, significantly reducing the false detection rate in such videos by the cue-based method. Despite great improvements compared with other methods, the false alarms per minute are still too high to apply in forensic science. Furthermore, the reported failure cases indicate that the method could not detect frame drop when the scene has almost no changes between adjacent frames. 

Long et al. [9] developed a coarse-to-fine deep convolutional neural network for duplication detection. First, an I3D network finds coarse-level suspected matches by using a sequence-to-sequence distance matrix, which narrows the search space. Then, a ResNet network [28] with 152 layers evaluates the likelihood using both video-level and frame-level scores. The duplicated frame inserted into the video usually yields artifacts due to inconsistency, which serves to locate the duplication in the whole video. This is an effective way to detect copied and inserted frames rather than deleted frames. Shanableh [39] proposed a novel approach to detect frame deletion by using machine learning techniques. The extracted features include prediction residuals, percentage of intra-coded macroblocks, quantization scales, and reconstruction quality. The method is suitable for both constant bit rate (CBR) and variable bit rate (VBR), whereas other learning-based solutions assume that forged videos are coded by CBR. On average, the experimental results showed a true positive rate of around 95% and a false negative rate of 4% as long as the number of deleted frames was not a multiple of the length of a GOP. 

Bakas et al. [10] proposed a novel approach for inter-frame video forgery detection based on 3D-CNN. Since inter-frame forgery occurs in the temporal domain, feeding raw image pixels directly did not yield excellent performance. A pixel-wise difference layer was introduced in the model at the beginning. Its accuracy reached 97% in the UCF-101 database, which was higher compared to the method described by Long et al. [9]. The performance of CNN models is found to be highly dependent on the training and test database, while the UCF-101 database was originally created for human action recognition. So, the validity of frame deletion in static scenes is still doubtful. Gowda et al. [14] also designed a 3D-CNN model for detecting and locating inter-frame forgery using a multi-scale structural similarity measurement algorithm. An absolute difference algorithm was introduced to minimize the temporal redundancy and identify forgery traces. 

Yang et al. [11] adopted an ingenious algorithm to detect and locate object removal in videos based on a network model. Firstly, a spatiotemporal trident network is introduced to output different high-frequency residual signals. Five consecutive frames are set as an input unit with every three consecutive frames acting as one branch to obtain three branches. Then, bi-directional long short-term memory runs as a classifier to determine whether the branches are pristine, forged, or hybrid frames. Compared with other network models, the loss function is designed as a combination of focal loss [55] and cross-entropy, while feature maps generated by C3D are used for classification. They reported successful detection and location accuracy of more than 96%. 

Fadl et al. [12] proposed the use of 2D-CNN spatiotemporal features and fusion for forgery detection. The first step involves spatiotemporal average fusion to extract features efficiently. In addition, the structural similarity index is applied to produce deep features of the entire video. A multi-class SVM is used as a classifier after feature fusion. The proposed system was found to perform better than hand-crafted feature approaches [32,46] on deletion, insertion, and duplication forgery. However, training videos indicate that the method works well when moving vectors can be detected in the video. Kaur et al. [13] expanded CNN for graphics forgery detection in video. The network could detect forged frames and point out forged regions. It reached an average accuracy of 98% when working with YouTube-compressed videos. However, the test videos were highly compressed and always contained moving objects. Girish et al. [15] adopted a long short-term memory (LSTM) model to detect region duplication. Spatiotemporal averaging is carried out to obtain background information. Next, feature extraction is performed using a GoogLeNet model. Then, a multi-subspace randomization and collaboration approach is used to select discriminative feature vectors. Finally, LSTM is applied for forgery detection. Patel et al. [56] adopted a multi-feature extraction framework optimized by CNN for use in inter-frame forgery detection. Speeded-up robust features (SURFs), HOG, mode-based fast digit features (MBFDFs), prediction residual gradient (PRG), and optical flow gradient (OFG) are used to train the forgery detection CNN model. Paing et al. [57] introduced a frame deletion system using an unsupervised learning algorithm. The hamming distance and Poisson probability mass function (PMF) distance are used to assess cross-correlations based on the Sobel edge operator. In general, the performance of network-based methods highly depends on the training set. The results obtained by the above methods were not proven to be effective under integer GOP deletion attacks in static scenes.

### 2.4. Double Encoding Detection Methods

Double encoding detection is another way to check a video. As double compression detection models have been successful for JPEG format [58,59], similar techniques have been established to detect double encoding in videos. Jiang et al. [37] proposed a method for the detection of double compression based on Markov statistics to authenticate video. A tampered video will be re-encoded and re-compressed after interpolation, so the double compression could be regarded as a questionable point in the video. Different from single images, each GOP is treated as a detection unit, which will be classified as singly or doubly compressed by Fisher’s linear discriminate analysis [60]. However, in practice, videos can be re-compressed during various transmissions yet not be forged.

Gironi et al. [37] also proposed a method to recognize the insertion and deletion of whole frames by double encoding detection. Variations in the prediction footprint (VPF) constitute a valid tool for double encoding detection [41]. It was observed that when an intra-frame was re-encoded as a predictive frame, an anomalous phenomenon occurred in the number of macroblocks (MBs). For a video encoded using a fixed size G1 for the first encoding and a fixed size G2 for the second encoding, periodic peaks were found in VPF, and the size of G1 could be estimated by the model. Gironi et al. [37] made a significant contribution by increasing the robustness of VPF, and the removal localization could be determined by iterative analysis. It must be pointed out that the method cannot detect frame manipulations when the attacker removes a whole GOP. 

Su et al. [61] argued that P frames, previously coded as I frames, will miss more high-frequency energy after the second compression. Inconsistencies in the energy of high-frequency components in neighboring P frames indicate deletion operation. As reported in that paper, the method applies only to MPEG-2 code. Even if a double encoding operation is detected, it can be regarded as a clue related to tampering, not proof. Notable work on the detection of forged videos in forensics was carried out by Wang et al. [35] in 2006. The video sequence is made up of a GOP structure. In one GOP, there are three types of frames: intra (I), predictive (P), and bi-directional (B). P frames are encoded with respect to the previous I or P frames, so the P frames in the same GOP are strongly correlated by referring to the same I frame directly or indirectly. Removing frames could lead to the P frame periodically referring to a different I frame after recoding. Meanwhile, a periodic increase in motion error occurs for every number of frame deletions, which will produce abnormal signals in the Fourier transform. 

Stamm et al. [36] modified the model of Wang et al. [35] by improving the robustness of the anti-forensics technique. To remove frame deletion fingerprints, the prediction error could be increased by purposefully choosing motion vectors without introducing unacceptable distortion. The baseline increase causes the spikes in the frequency domain curve to become gentle. Despite this, such an anti-forensics operation can leave behind new fingerprints in the video. A forensics technique to detect discrepancies between motion vectors stored in anti-forensics modified video and true motion vectors of that video scene is also proposed in [36]. Based on the model of Wang et al. [35], Liu et al. [40] introduced a sequence of the average residual of P frames (SAPR) to classify tampered and original content in H.264 video. The frequency domain transformed from SAPR showed that peaks also existed in the frequency domain for the original H.264 video. So, compared with the MPEG format, the peaks in the frequency domain do not necessarily suggest tampering in H.264 video. New and useful features are extracted to separate tampered and original videos by setting hard thresholds. 

Yu et al. [2] identified frame deletion by detecting abrupt changes in video streams. In theory, when a frame is re-encoded as a P frame into a new GOP, the correlation between this frame and its reference frame is weaker. Bakas et al. [44] further explored the detection method based on the GOP structure. Each frame is divided into macroblocks: intra-macroblock (I-MB), predicted macroblock (P-MB), and skipped macroblock (S-MB). P-MB can be further classified into three types based on motion vectors. Both the variation of macroblock features in the P frames and the variation of forward and backward motion vectors in the B frames of the video are taken into account to locate forged frames. Note that the mentioned deletion frame detection techniques based on GOP structure are useless when multiple GOPs are removed from the video.

As emphasized in Section 1, some approaches [1,33,34] should be highlighted owing to their anti-forensics detection performance. Zhang et al. proposed frame insertion and deletion detection based on inconsistencies in the LBP [1]. LBP was originally designed for texture analysis with sensitivity to subtle changes. LBP correlation coefficients (CCoLBP) are calculated to evaluate the correlation between successive frames. Abnormal CCoLBP points indicate frame insertion or deletion. The accuracy rate of the proposed method was reported to be 98.6% for 25 frames of insertion detection and 89.3% for 25 frames of deletion detection when applied to nearly 600 video databases. 

Zhao et al. introduced mutual information to video tamper detection [33]. A multi-scale normalized mutual information (MNMI) descriptor is introduced to characterize inter-frame similarity. Based on information theory, the mutual information of two events can describe the content similarity of adjacent frames. In order to decrease spatial sampling density with increased smoothing, three-layer Gaussian pyramids are chosen as filters. Expanding on the work of Wang [46] and Wu [62], the authors used 80 videos and diverse scenes in the experiments. The MNMI algorithm achieved 97.5% accuracy in detecting deletion and 82.1% in detecting localization. 

Bakas et al. proposed an interesting approach to detect and locate inter-frame forgeries based on inconsistencies in Haralick coded correlations [34]. A gray-level co-occurrence matrix (GLCM) is computed to provide a tabulation of frequencies of different combinations of pixel gray values. Then, 14 Haralick features based on GLCM are extracted to evaluate correlations with succeeding frames. Similar to LBP, GLCM is a classic method of texture feature extraction. GLCM performs well in structural information extraction, while LBP specializes at the pixel level. The experimental results proved that the proposed method outperformed other methods with an average overall accuracy of 0.97. Specifically, the proposed method is completely independent of GOP length, whereas multiple GOP length deletions limit most state-of-the-art detection approaches.

Digital videos are usually considered to be intuitive and reliable sources of evidence, as recorded videos are visible, and removing objects from videos is hard to do [4]. On the other hand, digital videos can be distorted by simply removing some key frames or replacing some frames with scenes captured by the same device, especially in static scenes, which are common manipulations used in forging videos in forensics. Such operation avoids detection by most methods mentioned in the references above [5,8,9,10,12,18,19,20,21,26,27,29,30,31,38,48,52]. The frame deletion detection approaches mentioned in the references above [2,35,36,37,39,40,42,44,61] could find traces of frame removal in static scenes. However, all of these methods have a common flaw in that they cannot detect frame deletion when the attacker removes the integer length of the whole GOP. Removing the integer length of the whole GOP is a useful anti-forensic technique to modify a video, which is a great challenge for forensic science. This paper presents a novel approach based on noise transfer matrices to detect such anti-forensic operations. In this paper, we focus on “meaningful” frame deletion forgery. “Meaningful” signifies that tampered videos are edited without any trace. Static scenes and the removal of multiple GOP lengths are introduced in our video samples. Several mentioned approaches [1,33,34] were selected to compare with the approach in our paper, owing to their anti-forensics characteristics. New learning-based approaches [52,53] were also tested to evaluate our proposed method. In total, we examined 80 original and “meaningful” forged videos to evaluate the performance of detecting anti-forensic attacks.

## 3. Proposed Approach

Before introducing the proposed method, we must point out the particularity of the dataset tested in this paper. Integer-length GOP deletion in static scenes is the anti-forensics operation we hope to detect. To the best of our knowledge, there is no specialized dataset to verify such detection performance. Therefore, the experiment reported in Section 4 was conducted using our proposed dataset, which is described in detail in Section 4.

### 3.1. Methods

In this section, we present the operation of the proposed video tampering detection method in detail. As stated in Section 1, the inter-frame noise correlation between adjacent frames of a natural video is inherently high. The noise stream is continuous in the video. The global and local features are correlated in successive frames with relatively gentle changes in time series. The changing of noise features is caused by coupling between sensors, characteristics, defects of sensors, temperature and working conditions of the equipment, etc. Integer GOP deletion in a static scene is an effective operation to evade most existing detection approaches, but the noise correlation would still be adversely affected, leaving the trace to be detected. We propose an anti-forensic forgery detection method based on a noise transfer matrix with a pyramid structure and adaptive weight adjustment. The method presented in this paper can be summarized into three main steps: (1) extracting noise, (2) computing the noise transfer matrix, and (3) adjusting the weights of the transfer matrix. Figure 2 shows a flowchart of the proposed detection method.

#### 3.1.1. Noise Extraction

A color image is converted to grayscale before processing. For static scenes, the noise intensity can be calculated by the averaging technique using Equation (1), where $N_{x,y,m,n}$ is noise intensity in the position (x,y) of the n^th^ frame which belongs to m^th^ GOP. $I_{x,y,m,n}$ is the intensity in the position (x,y) of the n^th^ frame which belongs to m^th^ GOP. $\bar{I_{x,y,m}}$ is the average intensity in position (x,y) of the m^th^ GOP containing the n^th^ frame:(1)$$N_{x,y,m,n}=I_{x,y,m,n}-\bar{I_{x,y,m}}$$

As reported by Li et al. [51], extracting the noise intensity directly at the pixel level will not work well. We adopted a three-layer pyramid structure inspired by Zhao et al. [33] to extract noise considering both local and structural features. Firstly, an overlapped 16 × 9 window is used to extract local noise features, including the maximum, mean, and standard deviation. Then, no overlapped 2 × 2 window is applied twice in succession on the maximum noise map extracted in the previous layer. The corresponding noise map also extracts the maximum, mean, and standard deviation. The sizes of the three feature maps are 120 × 120, 60 × 60, and 30 × 30, respectively. A flowchart of the noise feature maps is shown in Figure 3. Various sizes of windows (32 × 18 and 64 × 36) were tested to optimize the window size. The results of parameter optimization experiments are described in Section 4.

#### 3.1.2. Noise Transfer Matrix Computation

After noise extraction, the noise transfer matrix is computed as shown in Equation (2). A schematic diagram is shown in Figure 4. Similar computations are conducted on the maximum, mean, and standard deviation values in each layer. A total of nine noise transfer matrices are computed to evaluate the distance between adjacent frames.
(2)$$C_{i,j,n}=\frac{N_{i,j,n}}{N_{i,n}}$$

In Equation (2), $C_{i,j,n}$ stands for the transfer probability, and $N_{i,j,n}$ stands for the amount of noise feature intensity transfer from i to j in the corresponding position in n^th^ and (n + 1)^th^ frames, and $N_{i,n}$ denotes the total amount of noise features with intensity i in the n^th^ frame. The distance $D_{n+1}$ between successive frames can be calculated using Equations (3)–(6), where $D_{n+1,max}$ represents the distance between maximum noise features extracted from the three-layer structure. A similar operation is adopted to compute the distance of mean and standard deviation noise. If the dimensions of two matrices are not equivalent, the smaller dimension is filled with element zero to keep the same dimension as the larger one. Peaks in the single noise feature could be caused by interference in the video, while peaks standing in multiple noise features are more likely caused by frame deletion; an example is shown in Figure 5. Black circles indicate the result of extracted noise features in frame deletion points, and the blue circle indicates occasional spikes extracted in a single noise feature.
(3)$$D_{n+1,max}=\sqrt{\sum_{i=1}^{i}\sum_{j=1}^{j}\sum_{t=1}^{3}{\left(C_{i,j,max,t,n}-C_{i,j,max,t,n+1}\right)}^{2}}$$
(4)$$D_{n+1,mean}=\sqrt{\sum_{i=1}^{i}\sum_{j=1}^{j}\sum_{t=1}^{3}{\left(C_{i,j,mean,t,n}-C_{i,j,mean,t,n+1}\right)}^{2}}$$
(5)$$D_{n+1,sd}=\sqrt{\sum_{i=1}^{i}\sum_{j=1}^{j}\sum_{t=1}^{3}{\left(C_{i,j,sd,t,n}-C_{i,j,sd,t,n+1}\right)}^{2}}$$
(6)$$D_{n+1}=\sqrt[{3}]{D_{n+1,max}\times D_{n+1,mean}\times D_{n+1,sd}}$$

The same distance value means differences in various dimensions. An example of changing dimensions in an entire video is shown in Figure 6. In short videos, it does not make sense for the dimension to change at a slow gradient. However, in hundreds of frames, the diversity of dimensions can reach 50%, which should not be overlooked, as reported by Li et al. [51]. The transfer matrix would be more like an identity matrix with stronger correlations in successive frames. Conversely, the transfer matrix would be more like a uniform matrix. To normalize extreme situations, the results computed by Equations (3)–(5) should be multiplied by a regulation coefficient, K (Equation (7)), to eliminate the effect of different scales. The n in the formula means the dimension of the transfer matrix. Because the dimension of the transfer matrix (*n*) is relatively large, the normalization process is simplified by multiplying by a coefficient (Equation (8)).
(7)$$K=\frac{n}{\sqrt{n-1}}$$
(8)$$K=\sqrt{n}$$

The correlation of noise between adjacent frames can be influenced by multiple factors. In successive frames, minor disturbances will reduce the continuity of noise to some extent. For integral GOP deletion, frames in different GOPs will become back-to-back connections, reducing the correlations between noise features. Two consecutive high signals would appear in the FDP, as explained in Figure 7. The product of noise correlation amplifies the difference between minor disturbances in the video and integral GOP removal. The final correlation of noise continuity can be enhanced by using Equation (9). The function ln() represents the operation of taking the natural logarithm. An example of such an enhanced operation is shown in Figure 8, where the black circles indicate the selection of genuine FDP and the blue circles indicate other peaks. It can be inferred that enhanced operation highlights the abnormal fluctuation of noise features caused by frame removal.
(9)$$D_{n,f}=\ln\left(D_{n-1}\times D_{n}\right)$$

#### 3.1.3. Adjusting Transfer Matrix Weights

Equations (3)–(5) determine that all transfer matrix elements have the same contribution. It is a simple assumption but not reasonable for some elements to change rapidly, even in successive frames; others will change slowly despite the removal of an entire GOP. In practical forensic cases, disputed digital data usually last tens of seconds to a few minutes without supplementary digital data. The weight modification should be practical and efficient.

Inspired by the above, we introduce a self-adaption module to adjust the weights of the transfer matrix. Elements that participate in weight modification are decided in the first step. Weight amendment is learned from successive frames and discontinuous frames. In this paper, we apply the length of the whole GOP (25 frames) to discontinuous frame contribution learning and weight adjustment. A sketch map is shown in Figure 9. The comparison experiments with various intervals are presented in Section 4. In general, weights related to broken points in the frame will be strengthened, and weights related to successive frames will be weakened. All weights are initialized to 1. First, elements of the transfer matrix, in both successive and discontinuous frames, are arranged in descending order by Equations (10) and (11):(10)$${Var}_{s,i,j}={\left(C_{i,j,t,n}-C_{i,j,t,n+1}\right)}^{2}$$
(11)$${Var}_{d,i,j}={\left(C_{i,j,t,n}-C_{i,j,t,n-25}\right)}^{2}$$

The top elements in the two sets represent the maximum variance in successive frames and discontinuous frames. If an element arrears in the single set, it will provide useful information for weight adjustment; it appears twice, and the opposite information is learned from descending order sets, which will confuse weight adjustment. Based on the above, Equation (12), inspired by information entropy described by Zhao et al. [33], is established to determine the weights participating in modification:(12)$$Entropy\left(N\right)=-\sum_{N=1}^{N}{Var}_{s,N}\times C_{i,j}\times \log_{2}C_{i,j}+\sum_{N=1}^{N}{Var}_{d,N}\times C_{i,j}\times \log_{2}C_{i,j}$$
where ${Var}_{s,N}$ and ${Var}_{d,N}$ represent the top N elements in the arranged sets, and $C_{i,j}$ is the corresponding transfer probability of noise feature. The extreme value of the above function determines the elements participating in weight adjustment. Then, the weights of elements selected by Equations (10) and (12) will be weakened by Equation (13), while the weights of elements selected by Equations (11) and (12) will be strengthened by Equation (14):(13)$${Weights}_{w,N}=-\frac{\delta }{GL}\times \log_{10}{\left(C_{i,j,N,n}-C_{i,j,N,n+1}\right)}^{2}+0.000001$$
(14)$${Weights}_{s,t}=\frac{\sum_{N=1}^{N}{Weights}_{w,N}}{\sum_{N=1}^{N}{\left(C_{i,j,N,n}-C_{i,j,N,n+1}\right)}^{2}}\times {\left(C_{i,j,t,n}-C_{i,j,t,n+1}\right)}^{2}$$

${Weights}_{w,N}$ represents the *N*th element in the set determined by Equation (10), *GL* represents the frame length participating in weight adjustment, $C_{i,j,N,n}$ represents the corresponding value in the transfer matrix, and δ represents the power of adjustment. ${Weights}_{s,t}$ represents the t^th^ element in the set determined by Equation (11), and $C_{i,j,t,n}$ is the corresponding value in the transfer matrix. The intensity of δ influences the performance of the weight adjustment module, according to comparison experiments conducted to optimize the parameter. The results are shown in Section 4. Finally, the noise features computed by Equations (3)–(5) are modified by the weight adjustment module using Equations (15)–(17), where ω represents the corresponding weight after adjustment:(15)$$D_{n+1,max}=\sqrt{\sum_{i=1}^{i}\sum_{j=1}^{j}\sum_{t=1}^{3}{\omega_{i,j,max,t}\times \left(C_{i,j,max,t,n}-C_{i,j,max,t,n+1}\right)}^{2}}$$
(16)$$D_{n+1,mean}=\sqrt{\sum_{i=1}^{i}\sum_{j=1}^{j}\sum_{t=1}^{3}{\omega_{i,j,mean,t}\times \left(C_{i,j,mean,t,n}-C_{i,j,mean,t,n+1}\right)}^{2}}$$
(17)$$D_{n+1,sd}=\sqrt{\sum_{i=1}^{i}\sum_{j=1}^{j}\sum_{t=1}^{3}{\omega_{i,j,sd,t}\times \left(C_{i,j,sd,t,n}-C_{i,j,sd,t,n+1}\right)}^{2}}$$

## 4. Experiments

In this section, first we introduce the dataset and coarse filter, and then we describe several experiments, including optimization parameter experiments, ablation experiments, and comparison experiments, that were conducted to verify and test the performance of the proposed method. Finally, some visual examples and discussion will be given at the end of the section.

### 4.1. Dataset

To our knowledge, there is no specialized video database for integer GOP deletion in static scene detection. This is a blind spot of public-reported research; meanwhile, it makes sense in forensic practice. Therefore, we created our own database by shooting 80 static scene videos. Analog and digital signals are the two main kinds of videos in practical use. So, test videos containing analog and digital signals have more stringency in court science. We recorded 40 samples with a Sony HVR-25C as analog signals saved on magnetic tape and 40 samples with a Sony NXCAM as digital signals saved in memory cards. All videos are 500 frames in length and consist of static scenes with a frame rate of 25 frames per second, resolution of 1920 × 1080 pixels, GOP length of 25, and other parameters set to default value. Half of the videos are original, and half are tampered with by means of 100 frame deletion (4 GOP) in the 250th frame. Example images from the videos are shown in Figure 10. All of the genuine FDPs could not be observed frame-by-frame directly even by experienced forensic identifiers.

### 4.2. Coarse Filter

Before presenting our experiment results, we will first describe the preliminary detection of outliers. As reported in previous work [1,33,34], I frames are encoded independently in each GOP without any reference frames. The correlation value will cause a significant drop between the I frame and the last frame of the previous GOP. This is a periodic effect caused by the GOP structure. Our concern is whether an abnormal reduction point occurs caused by the periodic effect or integral GOP deletion. We adopted Chebyshev’s inequality to filter massive useless data in the curve by Equations (18) and (19) for the results of the three compared methods mentioned in Section 4. Similar operations were conducted as in [1,34]. *T*_1_ and *T*_2_ represent the filter thresholds; values higher than the threshold will be filtered. $\mu_{1}$ and $\mu_{2}$ represent the mean values of correlation distribution and corresponding derived function of the given video. $\sigma_{1}$ and $\sigma_{2}$ represent the standard deviation of correlation distribution and the corresponding derived function of the given video. $p_{1}$ and $p_{2}$ represent the standard deviation from the mean. In our paper, the value is set to 1 as a coarse processing stage. Samples are shown in Figure 11 and Figure 12.
(18)$$T_{1}=\mu_{1}-p_{1}\times \sigma_{1}$$
(19)$$T_{2}=\mu_{2}-p_{2}\times \sigma_{2}$$

Filtered by Equations (18) and (19), a portion of points are selected for further calculation to evaluate the performance, as indicated by the blue circles in Figure 12. The mean and standard deviation of filtered points are computed to evaluate the detection performance of the method by using the receiver operating characteristic (ROC) curve. The true positive rate (TPR) and false positive rate (FPR) with variable standard deviation from the mean is used to draw the ROC curve. Numbers in tables represent the standard deviation between the detection of true or false positives and the mean computed by the above equations. In our paper, 0.5 times standard deviation is the interval in the ROC curve.

Different from the compared methods [1,33,34], in our paper, peak points indicate a low correlation between extracted noise features. A similar filter operation is performed by Equations (20) and (21) to extract points for further processing. *T*_3_ and *T*_4_ represent the filter thresholds; values lower than the thresholds are filtered. $\mu_{3}$ and $\mu_{4}$ represent the mean values of correlation distribution and corresponding derived function of the given video. $\sigma_{3}$ and $\sigma_{4}$ represent the standard deviation of correlation distribution and the corresponding derived function of the given video. $p_{3}$ and $p_{4}$ represent the standard deviation from the mean. In our paper, the value is set to 1 as a coarse processing stage. A sample is shown in Figure 13. It should be pointed out that an obvious trough was observed in the initial stage of recording video in the results for all proposed noise-based methods. This phenomenon can be explained by the unstable noise features at the beginning of the recording. So, the results of the first 100 frames were excluded from subsequent filtering. Numbers in tables are similar to those used to evaluate the compared methods. The ROC curve was used to evaluate the detection performance of the proposed method compared to other methods.
(20)$$T_{3}=\mu_{3}-p_{3}\times \sigma_{3}$$
(21)$$T_{4}=\mu_{4}-p_{4}\times \sigma_{4}$$

### 4.3. Parameter Optimization Experiments

Control variable experiments are conducted to determine the optimal parameters for our method based on the 80 videos mentioned in Section 4.1. Scenes 1–20 and scenes 41–60 are original videos; the remaining videos are integral GOPs with the static scenes carefully removed. The coarse filter mentioned in Section 4.2 is used to extract points for further analysis. Noise feature extraction windows of different sizes (16 × 9, 32 × 18, 64 × 36), GOPs with different lengths involved in weight adjustment (from 0.25 to 2 GOPs, increasing by intervals of 0.25 GOP), and weight adjustment of different strengths (from 1 to 8, increasing by intervals of 1) were used to test each combination to optimize the parameters in the proposed method. Table 1 shows comparison results and Figure 14 shows ROC curves. Numbers in the “true positive” and “false positive” columns of the table represent the standard deviation from the mean.

Figure 14 and Table 1 show the results for different extraction window sizes with the same length of GOP involved in weight adjustment (1 GOP) and the same strength of weight adjustment (3). The results indicate that a large extraction window size will lead to blunting of the ROC curve, especially on the left. With regard to the principle of no punishment in court for doubtful cases, the left curve speaks much louder than the right. So, we accept a noise feature extraction window of 16 × 9 in this paper. It could be explained that noise is a local feature of the image. The appropriate window size optimizes filtering performance; a larger size will lose too many details, narrowing the gap between successive and interrupted frames. The above phenomena generally occur with different combinations of GOP length and weight adjustment strength.

Figure 15 and Figure 16 and Table 2 show the results for GOPs of different lengths involved in weight adjustment with the same extraction window size (16 × 9) and the same strength of weight adjustment (3). The results indicate that better performance was achieved with 1 GOP, so we adopted 1 GOP for weight adjustment in our paper. The ROC curve decreases with reduced weight amendment length if shorter lengths are selected. It could be explained that the phase of the learned and current frames is variable. Moreover, a learning length that is too short means inadequate weight adjustment, which leads to worse performance. Conversely, if a longer length is selected, the ROC curve will drop entirely within a narrow margin with increasing weight amendment size. Despite tiny changes in the ROC curves, the results of individual cases (examples shown in Table 2) have obvious distinctions with variable weight amendment size. It could be explained that a too long learning length will cause an overfitting of weights, leading to polarized results in individual cases (examples shown in Table 2). Meanwhile, the phase of the learned and current frames is variable and also leads to worse performance. The above phenomena generally occur with different combinations of extracted window size and weight adjustment strength.

Table 3 shows the results of weight adjustment strength with the same extraction window size (16 × 9) and the same length of GOP involved in weight adjustment (1 GOP). Figure 17 and Figure 18, respectively, describe the results with two groups of control variables (16 × 9 with 1 GOP, and 16 × 9 with 2 GOP). If lower strength is selected, the performance is the best at δ=3 with 1 GOP; then, it declines with significant overfitting. If higher strength is selected, the performance is the best at δ=6 with 2 GOP; then, it declines with gentle overfitting. Between the two, the performance with low strength and shorter length beats that with higher strength and longer length. We adopted δ=3 as the weight adjustment strength in our paper. It could be explained that for weight learning, a length of 2 GOP would be too far away (50 frames) to learn noise features from the current frame, which would reduce the performance. Meanwhile, weight learning with more combined frames would lead to a weight matrix that is more uniform. So, the ROC curve will be entirely gentle with an obvious drop on the left, as demonstrated with δ=3 and δ=6. In addition, overfitting will also be gentle, as demonstrated with δ=4 and δ=8. The above phenomena generally occur with different combinations of extraction window size and GOP length involved in weight adjustment.

Based on the above experiments, we adopted a noise feature extraction window size of 16 × 9, a weight learning length of 1 GOP, and a strength coefficient of 3 for our experiments.

### 4.4. Ablation Experiments

Ablation experiments were also conducted to verify the contribution of the pyramid structure, weight learning module, and fusion performance. The dataset and coarse filter are, respectively, mentioned in Section 4.1 and Section 4.2. The individual detection results for each part and the combined results are shown in Table 4, and the corresponding ROC curves are shown in Figure 19. 

The results of the ablation experiment are shown in Figure 19. We can see that both the pyramid structure and the weight learning module contribute to the performance of frame deletion detection and work best in combination. Comparing their ROC curves, the weight learning module improved the TPR rapidly with a low FPR. Although the pyramid structure only achieved relatively small improvement, in the individual cases (examples shown in Table 4), abnormal peaks in original videos were suppressed by the pyramid structure. On the contrary, the weight learning module highlighted frame detection deletion points while increasing the probability of normal points turning into suspected peaks. Our proposed method combines the advantages of pyramid and weight learning. For the purposes of forensic science, the proposed method could detect considerable integral GOP deletion in static scenes with low risk.

### 4.5. Comparison Experiments

We quantitatively evaluated the contribution of the proposed method by conducting comparison experiments. Dataset and coarse filter are, respectively, mentioned in Section 4.1 and Section 4.2. Multiple GOPs were removed from the original videos of static scenes, which strongly oppose existing detection technology. Both up-to-date learning-based detection methods [14,15] and anti-forensics forgery detection methods [1,33,34] were used for comparison. Before presenting the results of our paper, we will briefly describe the comparison methods [1,14,15,33,34].

#### 4.5.1. LBP Method

LBP was originally designed for texture analysis. Because of its simplicity and efficacy in image processing and recognition, LBP has been applied in many research areas related to images. The definition of LBP is given by Equations (22) and (23):(22)$${LBP}_{P}=\sum_{p=0}^{P-1}s\left(g_{p}-g_{c}\right)2^{P}$$
(23)$$s\left(x\right)=\left\{\begin{array}{c} 1,x\geq 0\\ 0,x<0 \end{array}\right.$$

*P* denotes a neighborhood of *P* sampling points around the central pixel, which is always a 3 × 3 window, $g_{c}$ denotes the gray value of the central pixel, and $g_{p}$ denotes the gray value of the neighboring pixel. If $g_{p}$ is greater than $g_{c}$, the corresponding binary code is 1; otherwise, it is 0. Finally, these binary codes are transformed into a decimal number. Theoretically, the CCoLBP will drop obviously at the FDP. The LBP method proposed by Zhang et al. [1] was selected for comparison in our paper.

#### 4.5.2. MNMI Method

In information theory, a communication system consists of a source, a sink, and a channel. Redundancy is associated with the possibility of every symbol’s occurrence or uncertainty in the message. The mutual information of two events in information theory can be used to describe the content similarity of two adjacent frames. Inspired by this idea, the information between adjacent frames can be measured by joint entropy, as shown in Equation (24). The average mutual information is shown in Equation (25). Greater average mutual information means a higher similarity between adjacent frames. Due to the range of average mutual information not being between 0 and 1, Equation (26) should be normalized as shown in Equation (26):(24)$$H\left(F_{t},F_{t+1}\right)=-\sum_{i=0}^{L-1}\sum_{j=0}^{L-1}p\left(I_{i}\left(F_{t}\right),I_{j}\left(F_{t+1}\right)\right)\times lbp\left(I_{i}\left(F_{t}\right),I_{j}\left(F_{t+1}\right)\right)$$
(25)$$MI\left(F_{t},F_{t+1}\right)=H\left(F_{t}\right)+H\left(F_{t+1}\right)-H\left(F_{t},F_{t+1}\right)$$
(26)$$NMI\left(F_{t},F_{t+1}\right)=\left\{\begin{array}{c} 0,\;\;\;\;\;\;\;\;\;\;\;\;\;\;\;\;\;\;\;\;\;H\left(F_{t},F_{t+1}\right)=0\\ \frac{H\left(F_{t}\right)+H\left(F_{t+1}\right)}{2H\left(F_{t},F_{t+1}\right)},H\left(F_{t},F_{t+1}\right)\neq 0 \end{array}\right.$$
where $p\left(I_{i}\left(F_{t}\right),I_{j}\left(F_{t+1}\right)\right)$ denotes the possibility of a gray value pair $\left(I_{i},I_{j}\right)$ appearing on the corresponding pixel position of $F_{t}$ and $F_{t+1}$, and lb is a logarithm of 2. Because spatial analysis at different scales could extract multi-level details of the image, three-layer Gaussian pyramids were adopted in the algorithm. The final MNMI result can be computed using Equation (27). The weights of the pyramid layers are $\omega_{0}$= 0.5, 0.286, 0.143, and 0.071. As a result, discontinuity points caused by inter-frame forgery will be highlighted in the MNMI. The MNMI method proposed by Zhao et al. [33] was selected for comparison in our paper.
(27)$$MNMI\left(t\right)=\sum_{k=0}^{3}\omega_{k}NMI\left(F_{t}\left(k\right),F_{t+1}\left(k\right)\right)$$

#### 4.5.3. Haralick Coded Method

Texture is one of the most important characteristics of an image. Haralick coding can extract structural information from images, while LBP focuses on pixel differences. In order to compute Haralick features, GLCM should be determined first, which provides a tabulation of frequencies of different combinations of pixel gray values in the image. Four GLCMs were constructed using Equations (28)–(31):(28)$$P_{0{}^{\circ},d}\left(a,b\right)=\left|\left\{\left(\left(k,l\right),\left(m,n\right)\right):k=m,\left|l-n\right|=d\right\}\right|$$
(29)$$P_{45{}^{\circ},d}\left(a,b\right)=\left|\left\{\left(\left(k,l\right),\left(m,n\right)\right):k-m=n-l=d,\;or-d\right\}\right|$$
(30)$$P_{90{}^{\circ},d}\left(a,b\right)=\left|\left\{\left(\left(k,l\right),\left(m,n\right)\right):\left|k-m\right|=d,l=n\right\}\right|$$
(31)$$P_{135{}^{\circ},d}\left(a,b\right)=\left|\left\{\left(\left(k,l\right),\left(m,n\right)\right):k-m=l-n=d,\;or-d\right\}\right|$$
where $\left(k,l\right)$ and $\left(m,n\right)$ indicate the locations of pixels with gray levels a and b, representing set cardinality. After GLCMs are computed, the result is normalized using Equation (32) before Haralick feature extraction:(32)$$N^{,}\left(i,j\right)=\frac{N\left(i,j\right)}{\sum_{i=1}^{n}\sum_{j=1}^{n}N\left(i,j\right)}$$
where $N\left(i,j\right)$ represents the $\left(i,j\right)$ element of the non-normalized GLCM, and $N^{,}\left(i,j\right)$ represents the corresponding element after normalization. In our paper, the dimension of GLCM is set as 1920 × 1080. Haralick features are extracted from the normalized GLCM for each frame, including the angular second moment, contrast, sum of squares, correlation, sum average, sum entropy, sum variance, inverse difference moment, entropy, difference variance, difference entropy, information measure of correlation, and max correlation coefficient. Finally, correlations between adjacent frames are computed using Equation (33) based on the extracted Haralick features:(33)$$r_{i}=\frac{\sum_{j}\left(F_{i}\left(j\right)-\bar{F_{i}}\right)\times \left(F_{i+1}\left(j\right)-\bar{F_{i+1}}\right)}{\sqrt{\sum_{j}{\left(F_{i}\left(j\right)-\bar{F_{i}}\right)}^{2}\times \sum_{j}{\left(\left(F_{i+1}\left(j\right)-\bar{F_{i+1}}\right)\right)}^{2}}}$$
where $r_{i}$ represents the correlation between the *i*^th^ and (*i* + 1)^th^ frames, $F_{i}\left(j\right)$ represents the *j*^th^ Haralick feature of the *i*^th^ frame, and $\bar{F_{i}}$ denotes the average value of all Haralick features of the *i*^th^ frame. Abrupt changes in the correlation curve are visible at the editing point, including frame deletion, insertion, and duplication. The Haralick coded method proposed by Bakas et al. [34] was selected for comparison in our paper.

#### 4.5.4. MS-SSIM Method

Gowda et al. [14] designed an MS-SSIM method to detect inter-frame deletion in videos. To differentiate the frames from each other, an absolute difference layer is added at the beginning of the model. In addition, a group of frames is constructed to improve accuracy and reduce computations. Pixel-wise discrepancies between adjacent frames are computed using Equations (34) and (35). The inter-frame forgery detection model is followed by rectified linear units (ReLU), batch normalization (BN), Maxpooling3D, global average pooling, dense, and SoftMax.
(34)$$P_{f}\left(m,n\right)=\left|K_{f}\left(m,n\right)-K_{f+1}\left(m,n\right)\right|$$
(35)$$D_{k}\left(m,n\right)=\left\{\begin{array}{c} 1,\;if\;P_{f}\left(m,n\right)>f\\ 0,\;otherwise\; \end{array}\right.$$

In the learning stage, the UCF-101 dataset is split randomly at a ratio of 3:1 for training and testing the model. After that, multi-scale structural similarity index measurement (MS-SSIM) is adopted to locate inter-frame forgery points if they exist.

#### 4.5.5. UFS-MSRC Method

Girish et al. [15] introduced the UMS-MSRC algorithm and LSTM network to detect inter-frame forgery. The model comprises five major steps: dataset collection, data pre-processing, feature extraction, feature selection, and classification. The data pre-processing operation combines temporal and spatial information, which outputs the scene background with a pale appearance of movement vectors. In the feature extraction and selection stage, the GoogleNet model extracts 1024 feature vectors that are fed to the UFS-MSRC algorithm to select discriminative feature vectors. For the final result, the UFS-MSRC algorithm outputs overall features in ascending order based on k-nearest neighbor and Laplacian graphs developed in subspaces. Finally, selected optimal feature vectors are given as the input to the LSTM module for video forgery detection. The LSTM classifier’s parameters are set as follows: maximum epoch = 100, learning rate = 0.01, batch size = 40, hidden layers = 4, layer 1 = 100 units, layer 2 = 100 units, layer 3 = 125 units, and layer 4 = 100 units. The SULFA dataset is split randomly at a ratio of 4:1 for training and testing the LSTM model.

#### 4.5.6. Learning-Based Method Results

Despite the excellent effects achieved in forgery video detection with the contribution of newly developed learning-based methods, proof of their utility in forensic science is still weak. In our experiments, learning-based methods [14,15] failed to detect frame deletion points (FDPs) in all 40 edited videos. No FDP could be detected as the lowest similarity point in the entire video. Examples are shown in Figure 20, Figure 21 and Figure 22.

In the figures, FDPs are indicated by black circles, while the lowest similarity points in the videos are indicated by blue circles. The results indicate that the learning-based methods [14,15] were completely unable to detect the delicate forgeries in our experimental videos. The decrease in similarity, as a result of other factors in the video, would cover the FDPs. Further verification is not necessarily based on current results. The phenomenon widely exists in the generalization of learning-based methods [63]. The out-of-distribution significantly reduces the performance of learning-based methods [64]. It should be highlighted that although impressive detection accuracy is achieved by many learning-based methods, the datasets used to verify the effect, such as UCF-101, VIFFD, SULFA, etc., are quite different from those used in forensics practice. UCF-101 was originally used for human action recognition research, VIFFD is made up of scenes from daily life, and SULFA was designed for the forensic analysis of tampered videos by presenting lifelike scenarios. In these databases, continuous motion vectors and local differences existing in successive frames can definitely be detected. A 3D-CNN model or LSTM model can detect FDPs in analogous videos containing continuous motion vectors by learning-based extracted features. In our paper, static scenes are involved in frame deletion detection because this is a common operation criminals use to forge videos. The instability of the focal plane, tiny changes in light sources, and various interference factors that occur during recording will be detected as local minimums of similarity. For the entire video, genuine FDPs will be covered by other low similarity points, as mentioned above, if the video is long enough. As learning-based methods are highly content-related, their extracted features, designed models, and parameters will not perform well in practical work, especially in integer-length GOP deletion in static scenes. Furthermore, such anti-forensics detection operation also veers from a series of methods based on the double compression detection principle proposed by Wang et al. [35]. Examples in UCF-101, VIFFD, and SULFA are shown in Figure 23, Figure 24 and Figure 25.

#### 4.5.7. Results of Robustness to Anti-Forensics Operation 

Several representative detection methods reported to be robust to anti-forensics operations [1,33,34] were selected for comparison to evaluate the performance of the method proposed in this paper. Table 5 shows the detection results, and Figure 26 shows the ROC curves. Note that if the genuine FDP in the compared method is filtered by the coarse stage described in Section 4 or is larger than the mean computed in the coarse filtering operation, the number is recorded as 0.

The LBP method is a pixel-level detection method. The detection capability is fairly weak, as seen by the poor performance of the ROC curve in Figure 26, and some FDPs can be filtered by the coarse process stage, which is shown as zeros in the LBPT column of Table 5. Nonetheless, the results still indicate that this method can detect frame deletion points in some cases. Some examples are shown in Figure 27. Genuine frame deletion detection is indicated by black circles. As this is a content-related approach, similar unusual peaks are also detected in original videos, covering the performance of such a detection approach to some extent. Moreover, LBP is an overly sensitive detection method, and false positive detections are common in some cases. Some obvious examples are shown in Figure 28 and Figure 29. Figure 28 shows the typical influence of jitter in the focal plane, and Figure 29 shows the typical influence of noise instability caused by a larger underexposed region in the frame. Obvious false positive detections are indicated by black circles. As mentioned above, the LBP method cannot be fully applied in forensic science practice to detect integral GOP length frame removal in forged static scene videos.

Haralick coding is a local structure-level detection method. It consumes vast running time when handling large images. In our paper, it took much more running time with little improvement over the LBP method. Some successful detection examples are shown in Figure 30. Genuine frame deletion detection is indicated by black circles. Compared with the LBP method, the Haralick coded method blunts the sensitivity to distraction in videos (seen in Table 5). False positive detections still occur with the Haralick coded method. Some obvious examples are shown in Figure 31. Camera shake leads to false positive detection, as indicated by black circles. Similar to the LBP method, the Haralick coded method also cannot be used to detect integral GOP length frame removal in forged static scene videos.

The MNMI method was inspired by information theory. Significantly improved ROC curve performance was obtained compared with the LBP and Haralick coded methods. This method has improved the detection of FDPs while restraining the influence of interference factors. Some examples are shown in Figure 32. True positive detections are indicated by black circles. Although the false positive detection is controlled by the specific nature of the MNMI method, failures could still be found in the results. Figure 33 shows the typical influence of the twinkling of the light source. Obvious false positive detections are indicated by black circles. As mentioned above, the MNMI method has little positive application in forensic science for detecting integral GOP length frame removal in forged static scene videos.

### 4.6. Discussion

The proposed approach is based on noise transfer matrix analysis. We adopt a pyramid structure and weight learning module to detect delicate frame deletion in forged videos. As shown in Figure 26 and Table 5, the proposed approach improves the performance of the ROC curve and individual cases more than other approaches. The results reveal the following: (1)The LBP, Haralick and MNMI algorithms detect suspected frame deletion points more accurately than random. Even though the ROC curves demonstrate poor performance, there is no doubt that those methods still provide useful information. False positive detection chance represents the dominant quantity in the video. Even considering the periodic effect caused by the GOP structure, the number of possible false positive detection points is dozens of times that of genuine positive detection points. An intuitive example is shown in Figure 12. Points indicated by blue circles indicate participants of the genuine FDP points.(2)Our approach has improved performance in detecting delicate frame deletion (integral GOP deletion in a static scene) in video compared with other approaches [1,33,34]. From Table 5, we can identify multiple examples of genuine FDPs detected by our approach, while these are ignored by the other approaches. Our approach reaches a TPR of 0.4 with an FPR of 0. However, the compared methods do not reach higher than 0.05. Because of the rigorousness of forensic science, vague clues cannot be accepted as evidence. A high TPR without any dispute can make a vital contribution to court proceedings.(3)Our approach reduces the possibility of false positive detection caused by various types of interference in video generation. This is a key point in forensic science in terms of verifying the authenticity of a video. In practical work, many novel approaches have been discarded due to the problem of false positive detection. In court, a suspected false positive detection in an entire video could overturn the results of genuine FDP detection. Examples will be given, presenting the characteristics of the proposed approach and comparing the approaches directly.

Despite demonstrating significant improvements, our method still has limitations, which can direct future work: (1) A TPR with a low FPR did not reach a high level, limiting the forensic science application. (2) Noise features are highly content dependent; thus, more videos should be tested to further verify and improve the proposed method. (3) The proposed measure is specifically aimed at detecting integral GOP deletions in static scenes of forged videos. Combining it with other approaches would help to expand the scope of application.

Unobservable anomalies due to frame deletion could be verified by the proposed approach, which was not possible with the other approaches. Some examples are shown in Figure 34 and Figure 35. The genuine FDPs detected by the proposed approach are indicated by black circles. Furthermore, the proposed approach is more robust to multiple types of interference in videos than the other approaches. Some examples are shown in Figure 36. Genuine FDPs are detected by the proposed approach, while suspected signals might be highlighted by the other approaches such as false positives. Obvious false positive detections by the compared approaches are indicated by blue circles, while genuine frame deletion point detections by the proposed approach are indicated by black circles. Because of the content-dependent nature of the proposed approach, obvious changes in a video, such as twinkling of the light source, cannot be avoided, as with the compared approaches. An example is shown in Figure 37. Obvious false positive detections are indicated by black circles. As mentioned above, the proposed approach can speak much louder than the compared approaches in forensic science with regard to detecting integral GOP length frame removal in forged static scene videos.

## 5. Conclusions

Determining the authenticity of a video poses a considerable challenge for modifying it easily without a trace. In forensic science, frame deletion can have serious consequences, such as unjust sentencing by courts. Meanwhile, many videos are presented as evidence in court. Removing integral GOP length frames in a static scene is an effective method of forging videos that cannot be accurately detected using current methods. Furthermore, such operation is convenient for anyone with publicly available software as long as they have a certain understanding of current detection methods.

This paper presents a method for anti-forensic forgery video detection based on noise transfer matrix analysis. Compared to three other anti-forensic forgery detection methods and newly developed learning-based methods, the proposed method significantly improves the performance of the ROC curve, especially in the low FPR interval, which is highly relevant in forensic science.

## Figures and Tables

**Figure 1 sensors-24-05341-f001:**
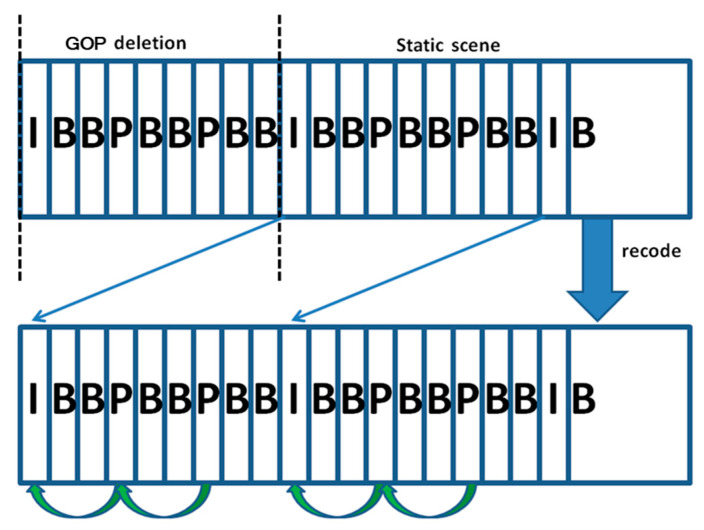
Sketch map of anti-forensics operation.

**Figure 2 sensors-24-05341-f002:**
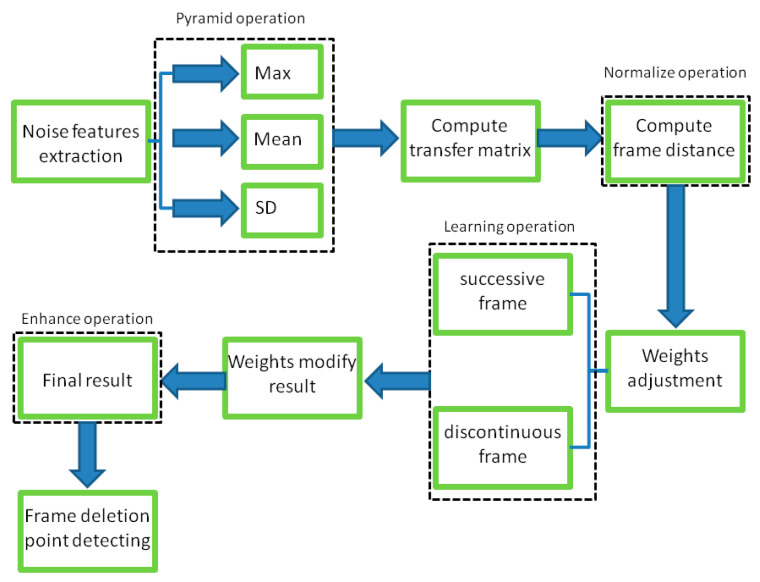
Flowchart of proposed detection method.

**Figure 3 sensors-24-05341-f003:**
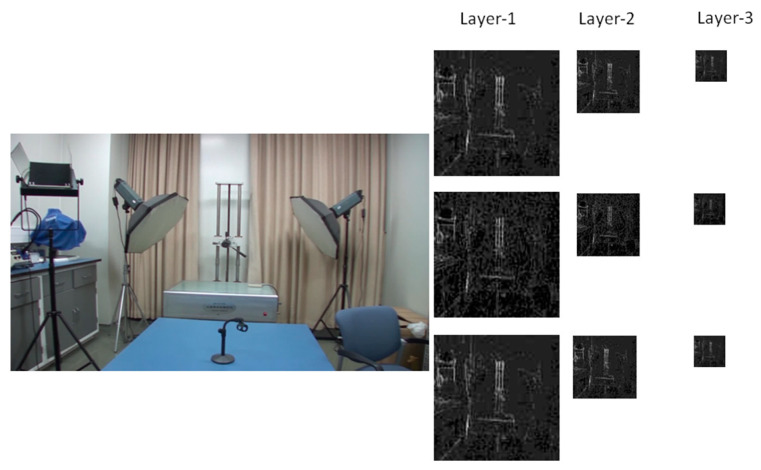
Flowchart of 3-layer noise feature maps.

**Figure 4 sensors-24-05341-f004:**
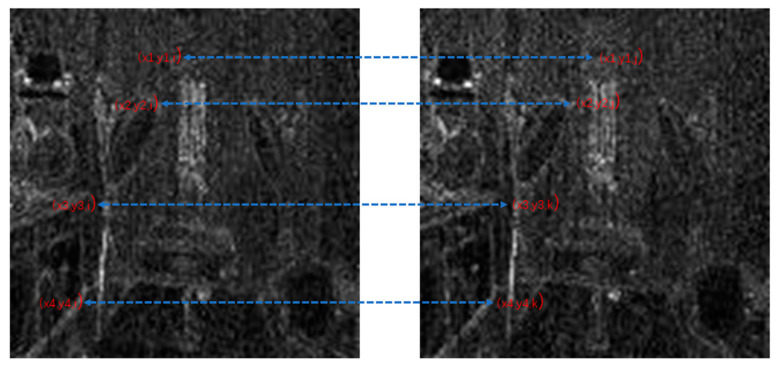
Schematic diagram of noise transfer matrix.

**Figure 5 sensors-24-05341-f005:**
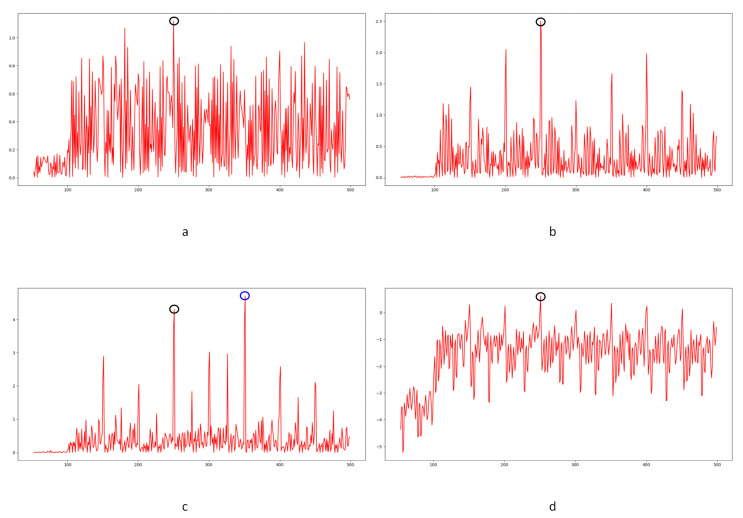
Example of multiple noise features extracted in scene 11: (**a**) maximum noise; (**b**) mean noise; (**c**) standard deviation noise; (**d**) combined result.

**Figure 6 sensors-24-05341-f006:**
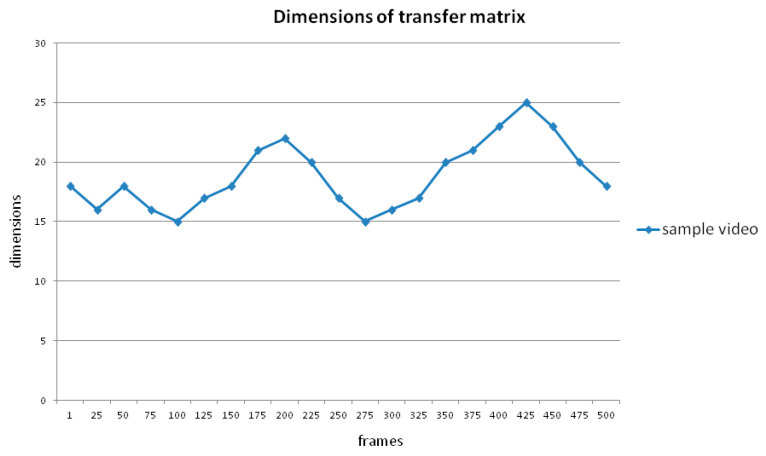
Schematic diagram of dimensions of transfer matrix in a sample video.

**Figure 7 sensors-24-05341-f007:**
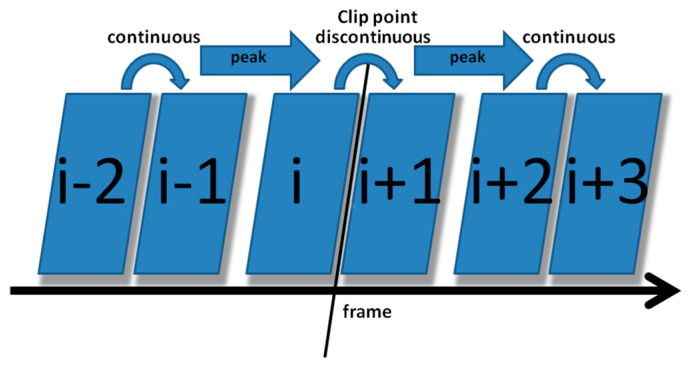
Diagram of two consecutive peaks appearing in FDP.

**Figure 8 sensors-24-05341-f008:**
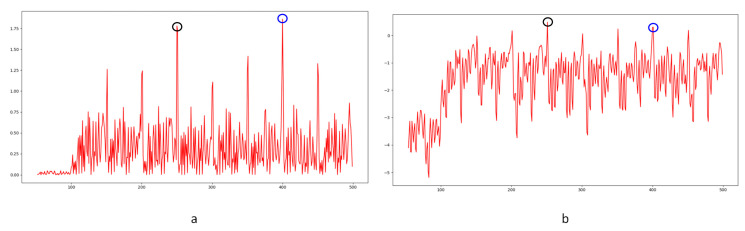
Example of enhanced operation in scene 79: (**a**) original result; (**b**) modified result from enhanced operation.

**Figure 9 sensors-24-05341-f009:**
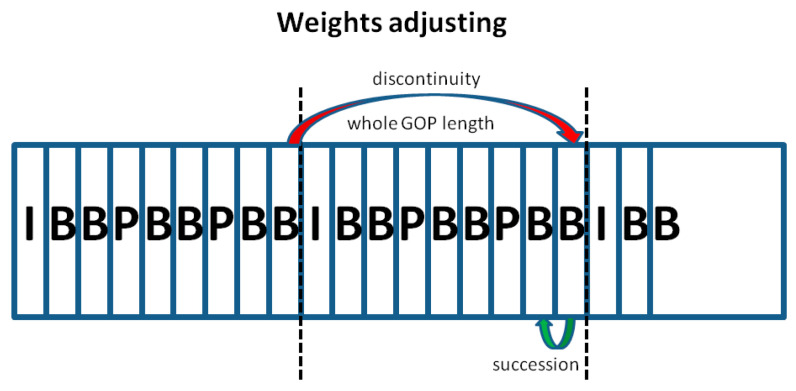
Sketch map of weight adjustment.

**Figure 10 sensors-24-05341-f010:**
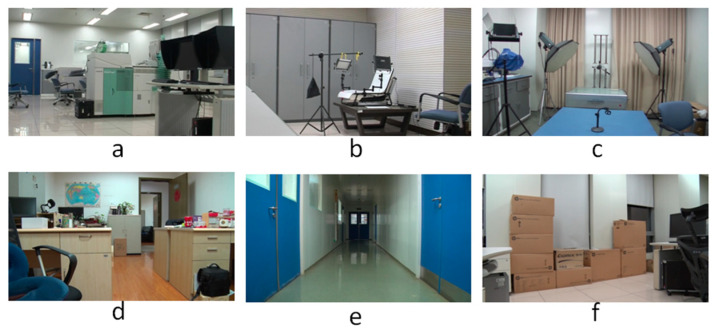
Examples of static scenes tested in this paper: (**a**) static scene 4; (**b**) static scene 8; (**c**) static scene 14; (**d**) static scene 22; (**e**) static scene 30; (**f**) static scene 37.

**Figure 11 sensors-24-05341-f011:**
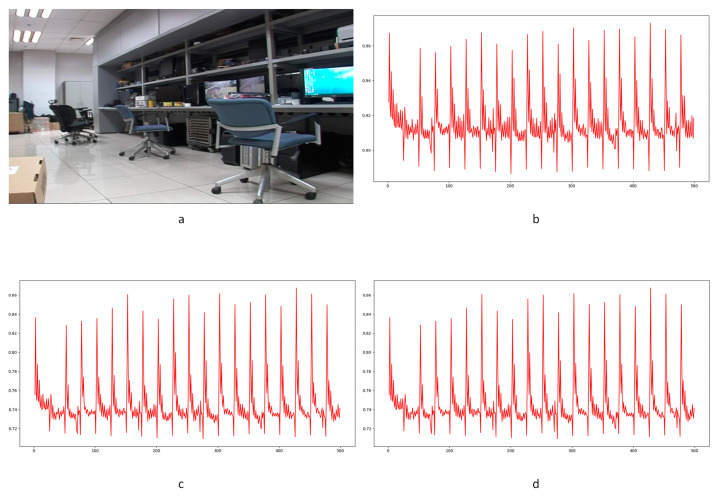
Results for scene 3: (**a**) original video; (**b**) result of LBP method; (**c**) result of MNMI method; (**d**) result of Haralick coded method.

**Figure 12 sensors-24-05341-f012:**
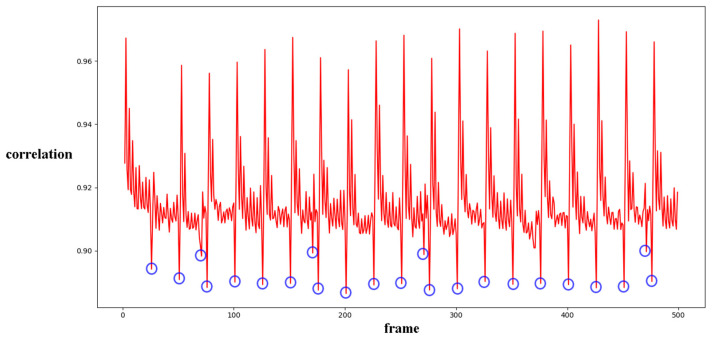
Filtered results of LBP method for scene 3 (blue circles).

**Figure 13 sensors-24-05341-f013:**
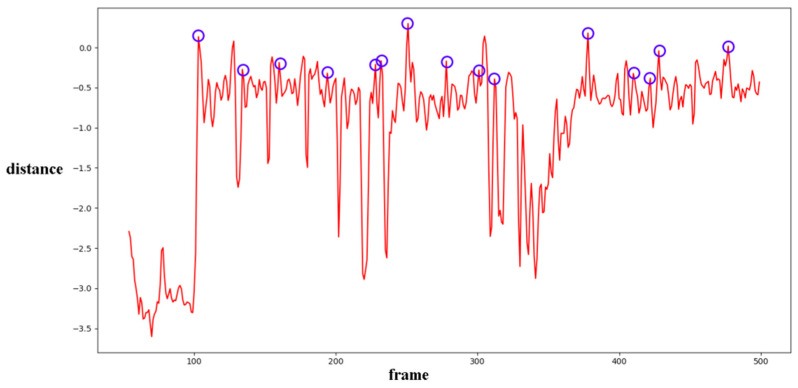
Filtered results of proposed method for scene 3 (blue circles).

**Figure 14 sensors-24-05341-f014:**
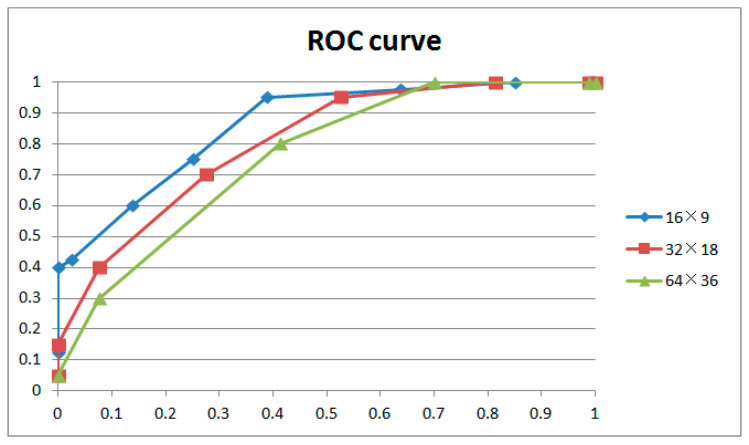
ROC curves for various extraction window sizes.

**Figure 15 sensors-24-05341-f015:**
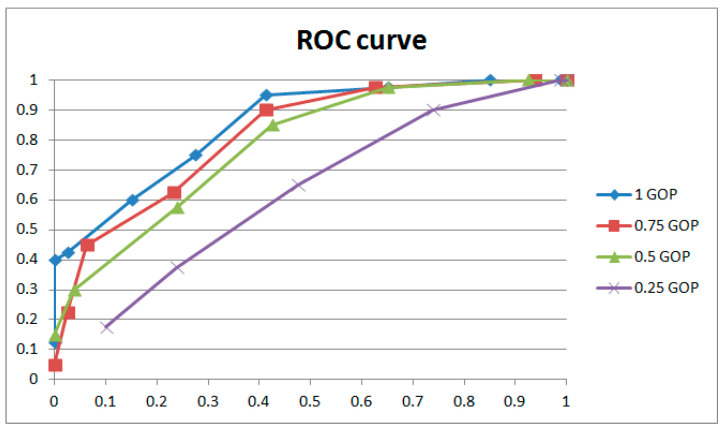
ROC curves of GOPs of various lengths involved in weight adjustment (1).

**Figure 16 sensors-24-05341-f016:**
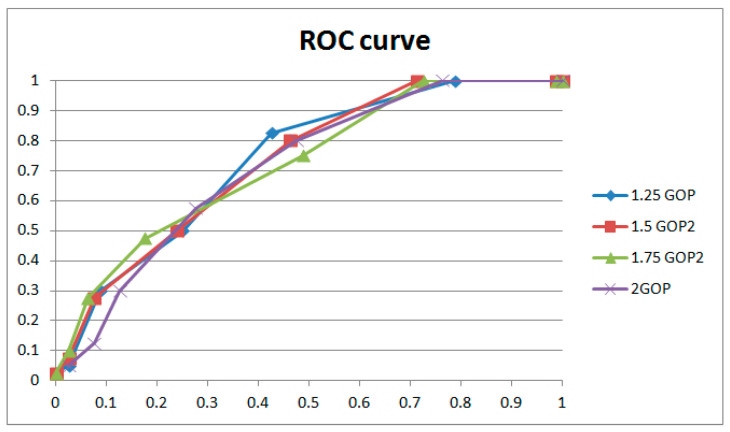
ROC curves of GOPs of various lengths involved in weight adjustment (2).

**Figure 17 sensors-24-05341-f017:**
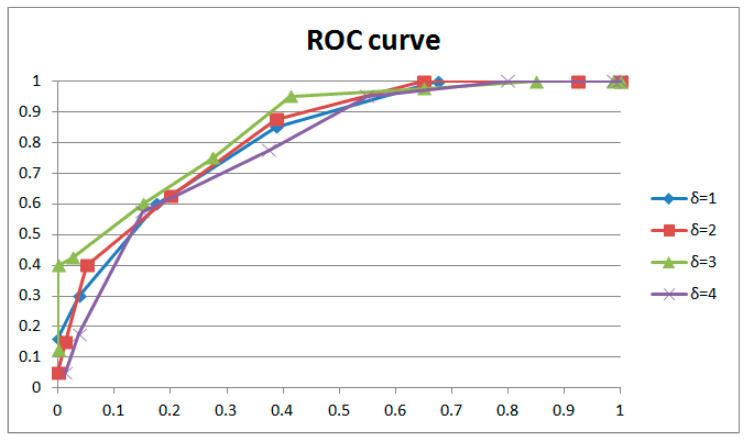
ROC curves for various weight adjustment strengths (1).

**Figure 18 sensors-24-05341-f018:**
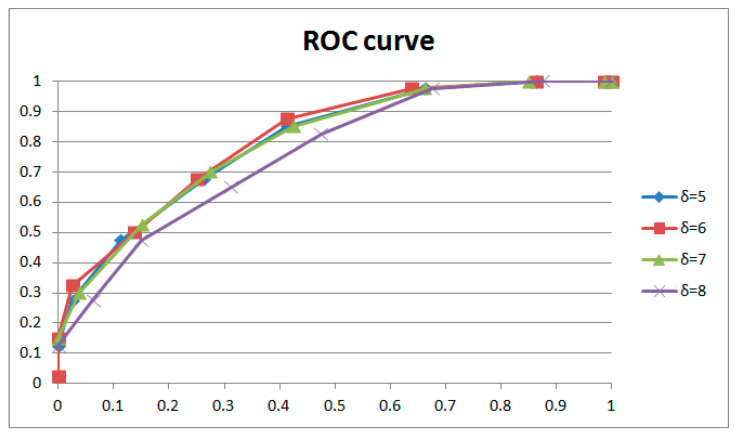
ROC curves for various weight adjustment strengths (2).

**Figure 19 sensors-24-05341-f019:**
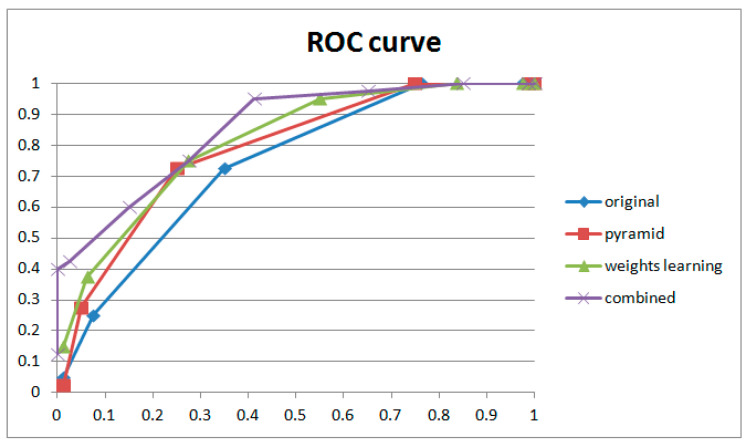
ROC curves for ablation experiments.

**Figure 20 sensors-24-05341-f020:**
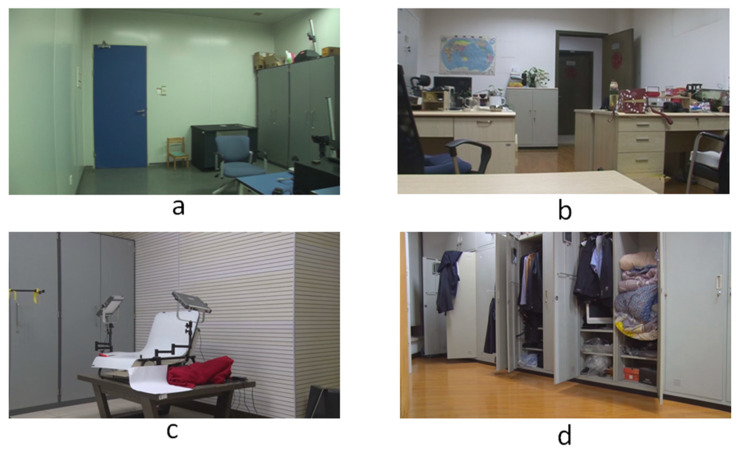
Example of detection scene with learning-based method: (**a**) static scene 21; (**b**) static scene 22; (**c**) static scene 68; (**d**) static scene 69.

**Figure 21 sensors-24-05341-f021:**
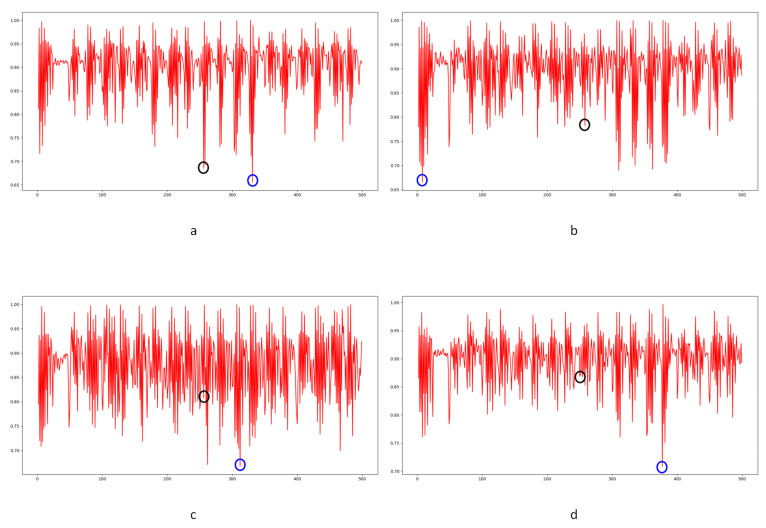
Example of detection results with MS-SSIM method: (**a**) static scene 21; (**b**) static scene 22; (**c**) static scene 68; (**d**) static scene 69.

**Figure 22 sensors-24-05341-f022:**
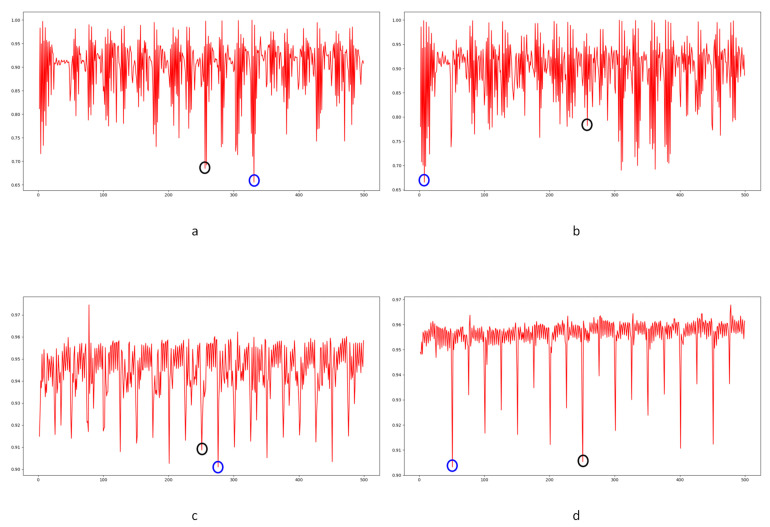
Example of detection results with UFS-MSRC method: (**a**) static scene 21; (**b**) static scene 22; (**c**) static scene 68; (**d**) static scene 69.

**Figure 23 sensors-24-05341-f023:**
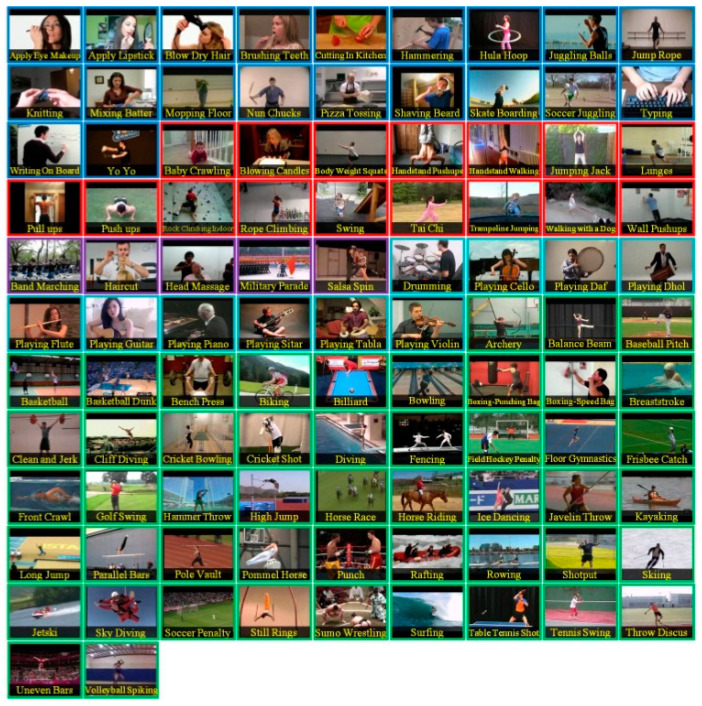
Video examples in UCF-101 database.

**Figure 24 sensors-24-05341-f024:**
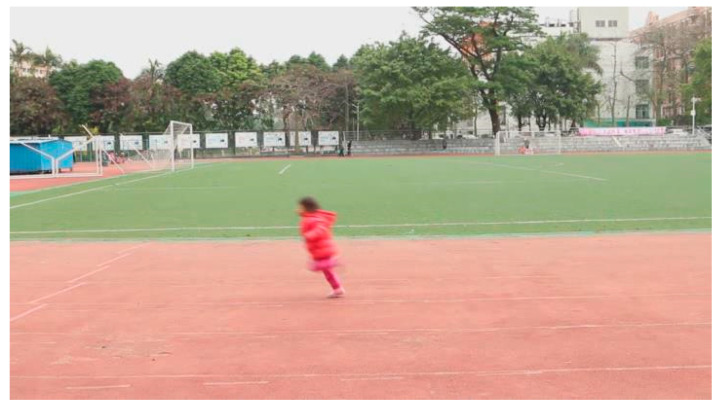
Video example in VIFFD database.

**Figure 25 sensors-24-05341-f025:**
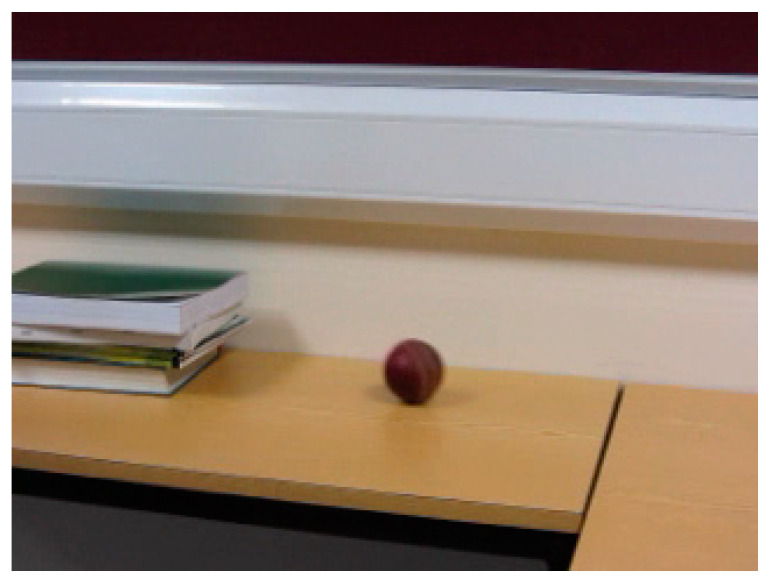
Video example in SULFA database.

**Figure 26 sensors-24-05341-f026:**
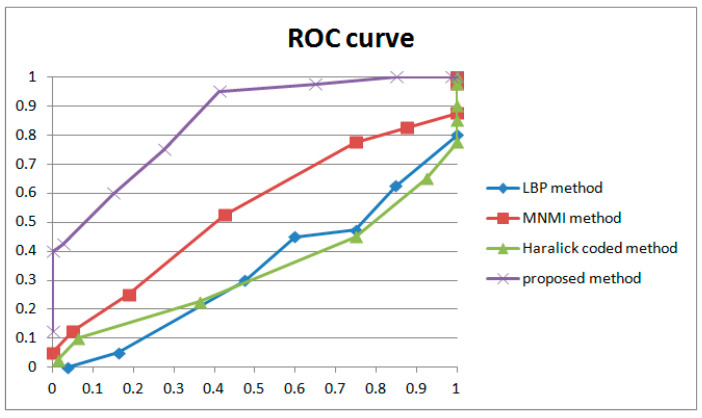
ROC curves of compared methods.

**Figure 27 sensors-24-05341-f027:**
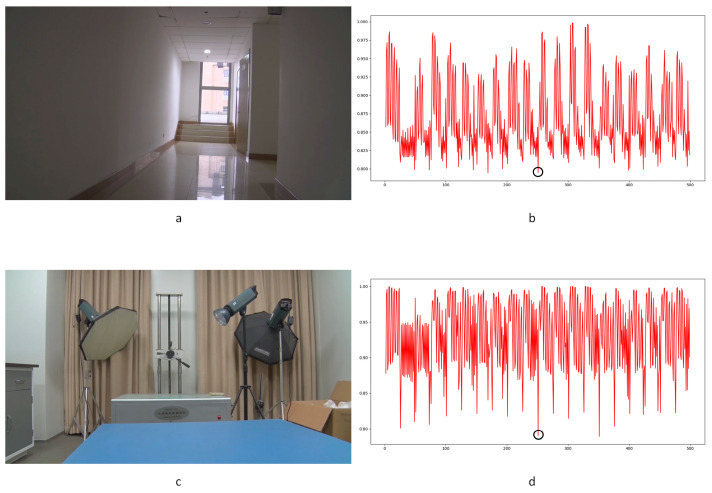
Examples of detection performance of LBP method: (**a**) static scene 73; (**b**) LBP result for static scene 73; (**c**) static scene 74; (**d**) LBP result for static scene 74.

**Figure 28 sensors-24-05341-f028:**
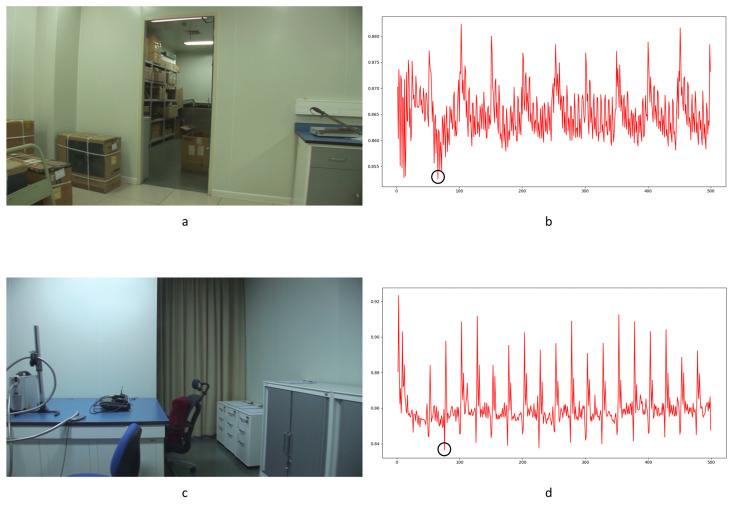
Examples of the typical influence of jitter in the focal plane: (**a**) static scene 5; (**b**) LBP result for static scene 5; (**c**) static scene 6; (**d**) LBP result for static scene 6.

**Figure 29 sensors-24-05341-f029:**
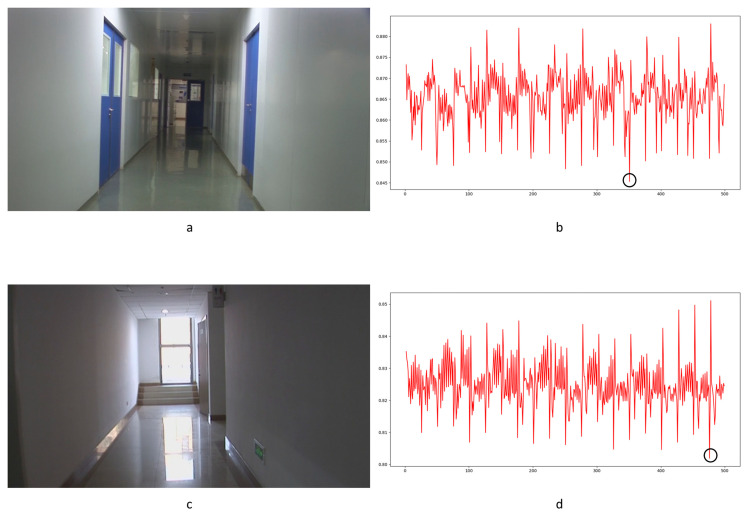
Examples of the typical influence of noise instability: (**a**) static scene 10; (**b**) LBP result for static scene 10; (**c**) static scene 13; (**d**) LBP result for static scene 13.

**Figure 30 sensors-24-05341-f030:**
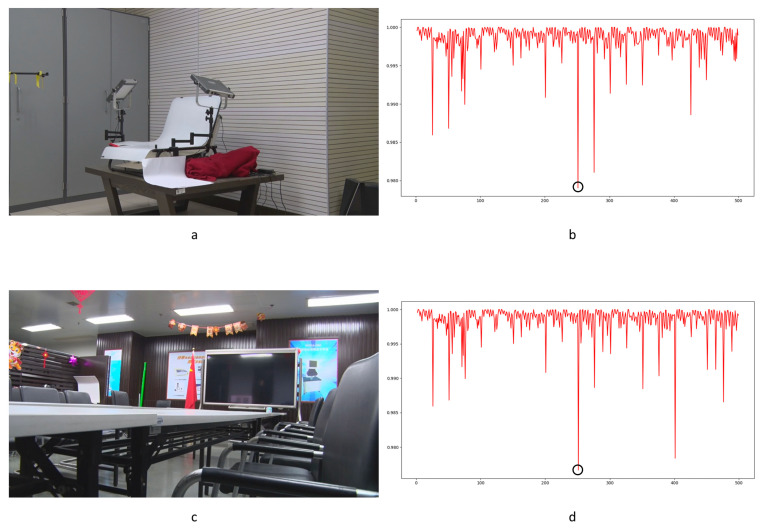
Examples of detection performance of Haralick coded method: (**a**) static scene 68; (**b**) Haralick coded result for static scene 68; (**c**) static scene 71; (**d**) Haralick coded result for static scene 71.

**Figure 31 sensors-24-05341-f031:**
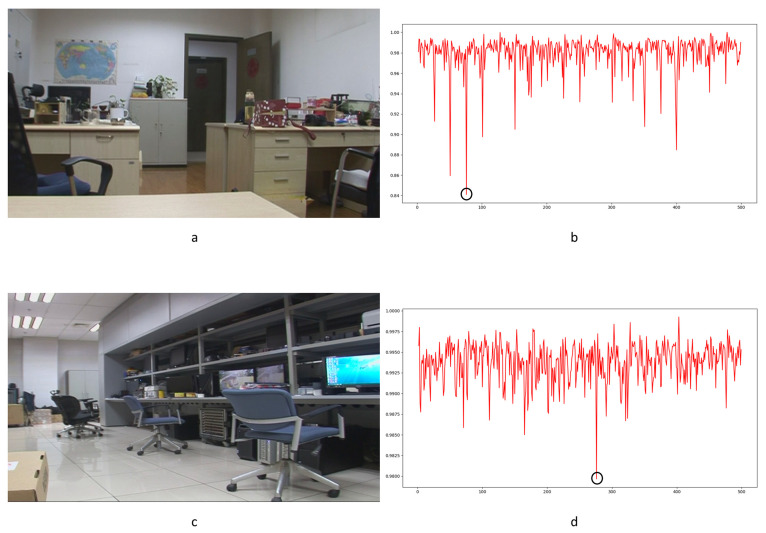
Examples of the typical influence of camera shake: (**a**) static scene 22; (**b**) Haralick coded result for static scene 23; (**c**) static scene 22; (**d**) Haralick coded result for static 23.

**Figure 32 sensors-24-05341-f032:**
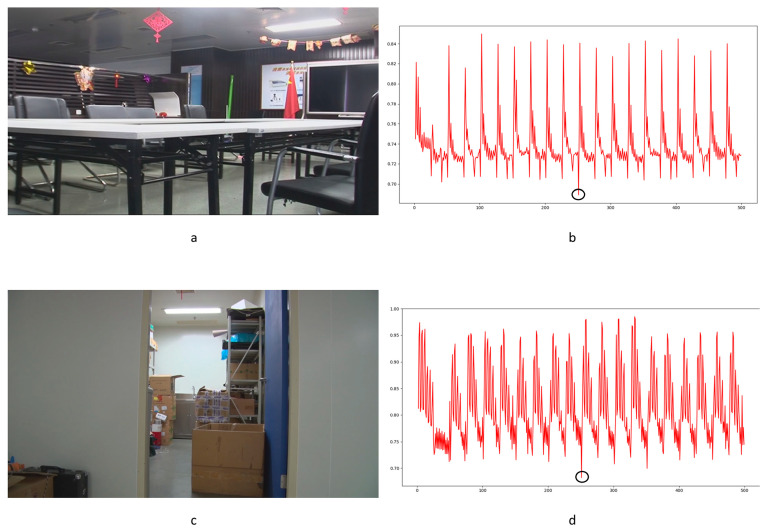
Examples of detection performance of MNMI method: (**a**) static scene 31; (**b**) MNMI result for static scene 31; (**c**) static scene 65; (**d**) MNMI result for static scene 65.

**Figure 33 sensors-24-05341-f033:**
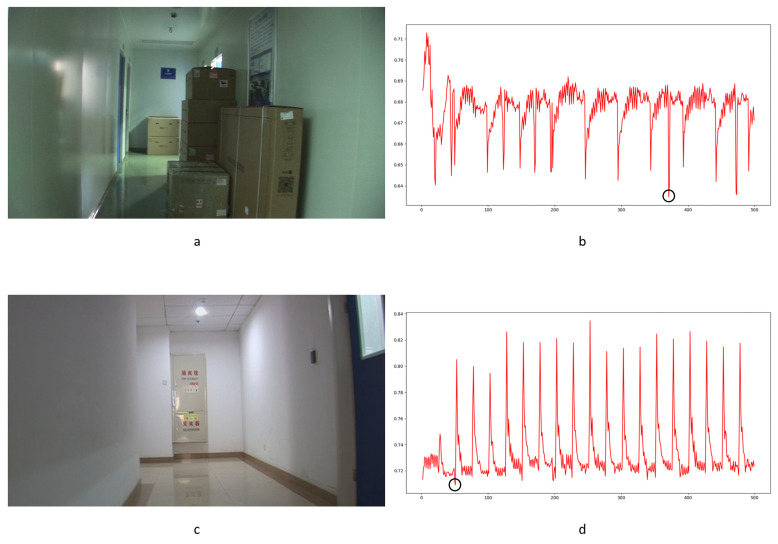
Examples of the typical influence of twinkling of light source: (**a**) static scene 37; (**b**) MNMI result for static scene 37; (**c**) static scene 38; (**d**) MNMI result for static scene 38.

**Figure 34 sensors-24-05341-f034:**
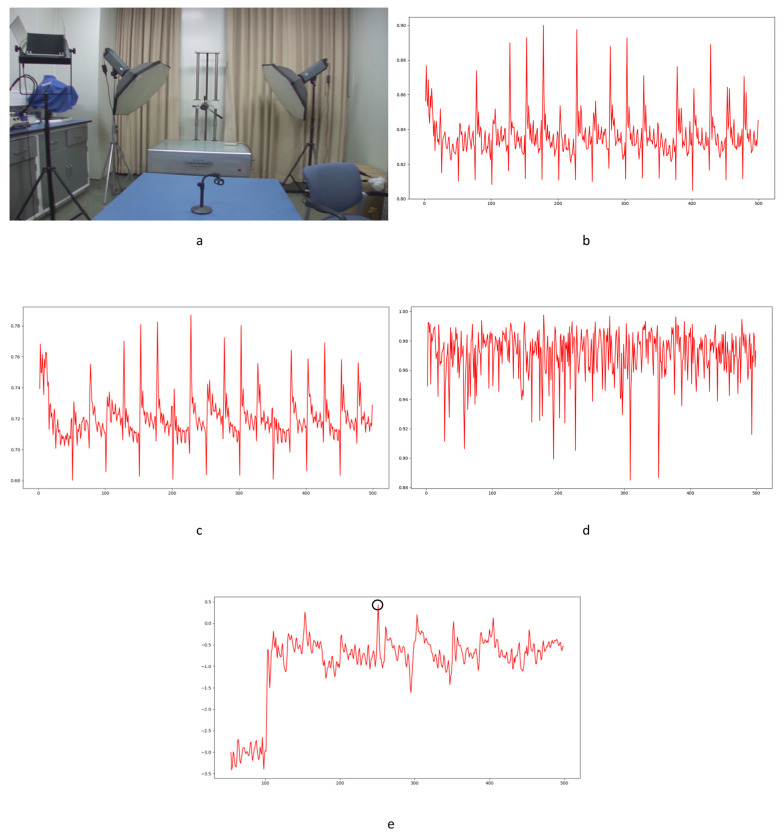
Examples of compared results: (**a**) static scene 34; (**b**) LBP method result for static scene 34; (**c**) MNMI method result for static scene 34; (**d**) Haralick coded method result for static scene 34; (**e**) proposed method result for static scene 34.

**Figure 35 sensors-24-05341-f035:**
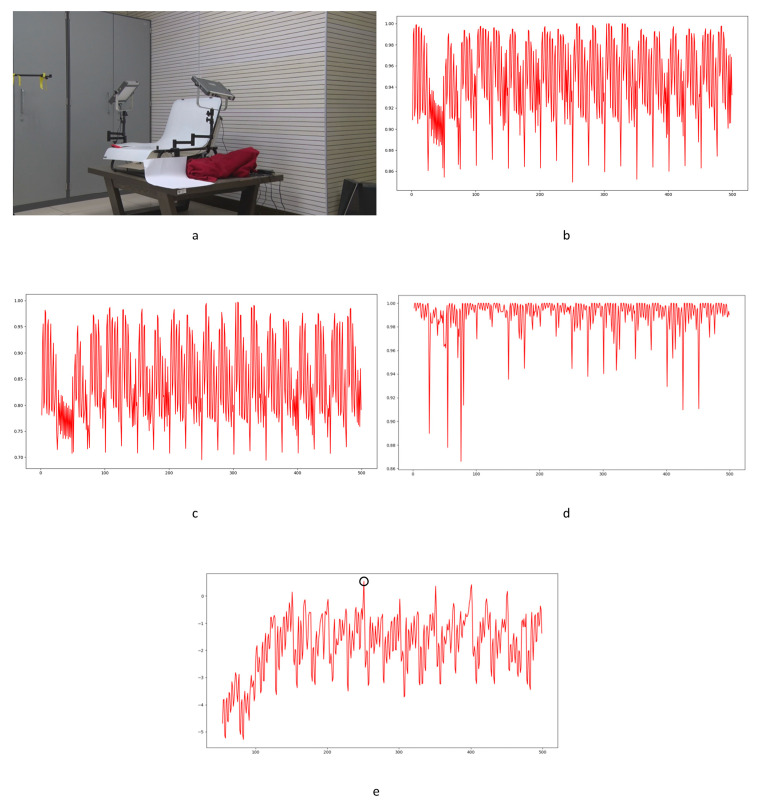
Examples of compared results: (**a**) static scene 68; (**b**) LBP method result for static scene 68; (**c**) MNMI method result for static scene 68; (**d**) Haralick coded method result for static scene 68; (**e**) proposed method result for static scene 68.

**Figure 36 sensors-24-05341-f036:**
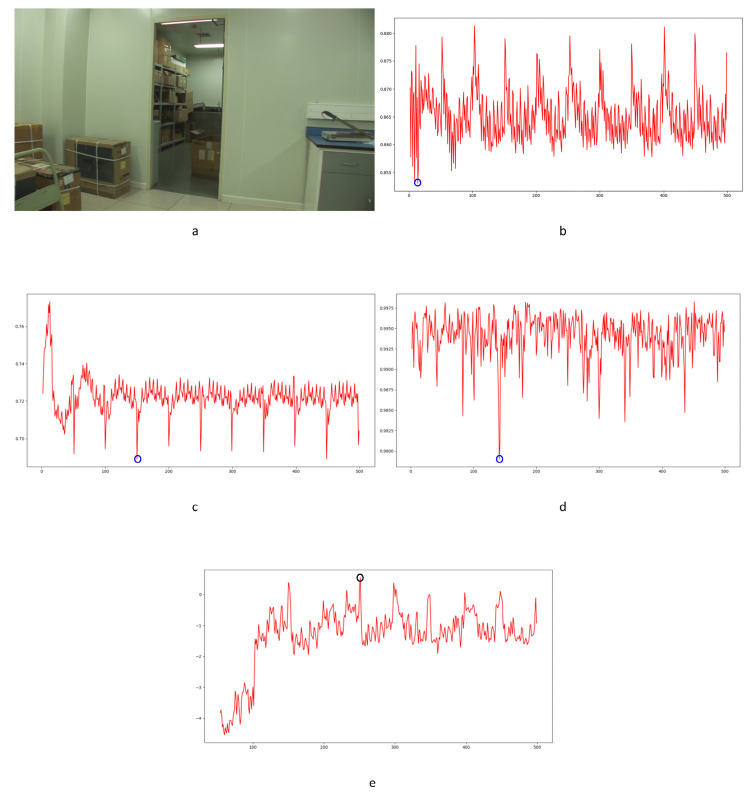
Examples of compared results: (**a**) static scene 25; (**b**) LBP method result for static scene 25; (**c**) MNMI method result for static scene 25; (**d**) Haralick coded method result for static scene 25; (**e**) proposed method result for static scene 25.

**Figure 37 sensors-24-05341-f037:**
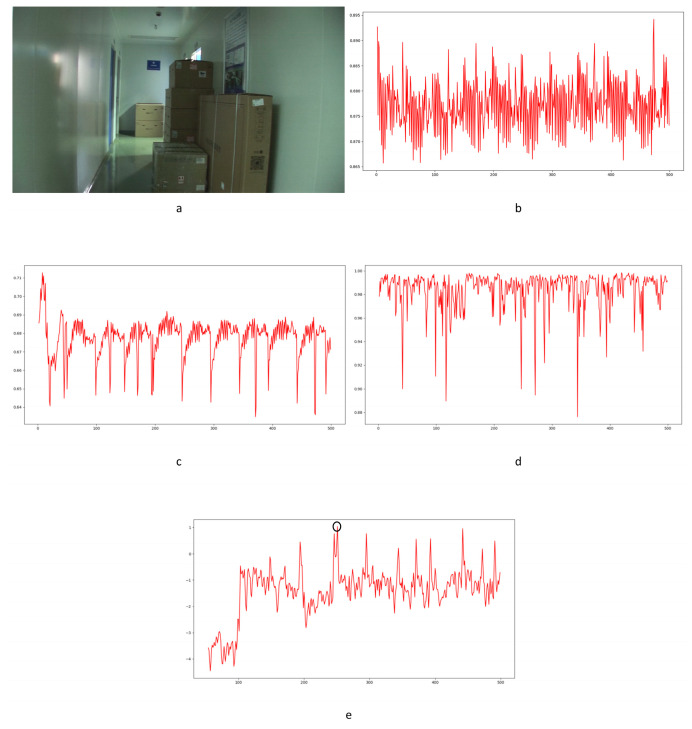
Examples of compared results: (**a**) static scene 37; (**b**) LBP method result for static scene 37; (**c**) MNMI method result for static scene 37; (**d**) Haralick coded method result for static scene 37; (**e**) proposed method result for static scene 37.

**Table 1 sensors-24-05341-t001:** Comparison results for various extraction window sizes.

Scene	16 × 9 True Positive	16 × 9 False Positive	32 × 18 True Positive	32 × 18 False Positive	64 × 36 True Positive	64 × 36 False Positive
1	/	1.5	/	1.5	/	1.5
2	/	2.5	/	2	/	2
3	/	2.5	/	2.5	/	2.5
4	/	3	/	3	/	2.5
5	/	2	/	2.5	/	2
6	/	2	/	2	/	2
7	/	2	/	2	/	2
8	/	2	/	2.5	/	2.5
9	/	2	/	2.5	/	2
10	/	3	/	2.5	/	2.5
11	/	2.5	/	2	/	2
12	/	3	/	3	/	2.5
13	/	2	/	2	/	2
14	/	2	/	2.5	/	2.5
15	/	2	/	2	/	2
16	/	2.5	/	2	/	2
17	/	2.5	/	2	/	2
18	/	2	/	2	/	2
19	/	2.5	/	2.5	/	2.5
20	/	3	/	2.5	/	2.5
21	4	2.5	3	2.5	2.5	2
22	3	3	3	3	3	3
23	4	3.5	3	2.5	2.5	2.5
24	3.5	3.5	3	2.5	2.5	2.5
25	4.5	3.5	3	3	3	3
26	3.5	2.5	2.5	2	2.5	2
27	5	3.5	3.5	3	3	2.5
28	3.5	3	3.5	3	3	3
29	4	4.5	3.5	3.5	3	3
30	4	4.5	3.5	3.5	3	3
31	6	4	4.5	3	3	3
32	5	3.5	4	3.5	3.5	3
33	5.5	3	4	3	3	3
34	5.5	3.5	4	3.5	3	3
35	4.5	4.5	4	4	3.5	3.5
36	3.5	3.5	3	3	2.5	2.5
37	4.5	4.5	4	4	3	3
38	4.5	3.5	3.5	3	3	3
39	4	3	3	3	3	3
40	3.5	5	3.5	3.5	3	3
41	/	2	/	2	/	2
42	/	2	/	2	/	2
43	/	2.5	/	2.5	/	2.5
44	/	2.5	/	2.5	/	2
45	/	2.5	/	2.5	/	2
46	/	2.5	/	2.5	/	2
47	/	3	/	2.5	/	2.5
48	/	3	/	2.5	/	2.5
49	/	2.5	/	2.5	/	2
50	/	2.5	/	2.5	/	2
51	/	3	/	2	/	2
52	/	3	/	3	/	2.5
53	/	4	/	2.5	/	2.5
54	/	3.5	/	3	/	2.5
55	/	2.5	/	2.5	/	2.5
56	/	3	/	3	/	3
57	/	4.5	/	3.5	/	3
58	/	2.5	/	2	/	2
59	/	3	/	2.5	/	2
60	/	2.5	/	2.5	/	2.5
61	5.5	4	4	4	3.5	3
62	5	4.5	3.5	3.5	3	3
63	5	5	3.5	3.5	3	3
64	5.5	4.5	5	4	3.5	3.5
65	6	4	4	3.5	4	3
66	4.5	3	3.5	3	3	3
67	5.5	4	4	3.5	3.5	3.5
68	5	3	3.5	3	3	2.5
69	4	3	3	3	3	3
70	4.5	3	3.5	3.5	3.5	3.5
71	3.5	3.5	3	3	2.5	2.5
72	6	4.5	4.5	4	3.5	3.5
73	6	4	4.5	3.5	3.5	3
74	4.5	4.5	4	4	3.5	3.5
75	3.5	3	3.5	3	3	3
76	3.5	3	3	3	2.5	2.5
77	2.5	4	2.5	3.5	2.5	3
78	6	3.5	5	3.5	4	3
79	5.5	4	4	3.5	3.5	3
80	5	4	4.5	3.5	3	3

**Table 2 sensors-24-05341-t002:** Comparison results of variable weight adjustment and length.

Scene	0.75 True Positive	0.75 False Positive	1 True Positive	1 False Positive	1.25 True Positive	1.25 False Positive
1	/	2	/	1.5	/	2.5
2	/	2.5	/	2.5	/	2.5
3	/	2.5	/	2.5	/	3
4	/	2.5	/	3	/	3
5	/	2.5	/	2	/	2.5
6	/	2	/	2	/	2.5
7	/	2	/	2	/	3
8	/	2.5	/	2	/	2.5
9	/	2.5	/	2	/	3
10	/	2.5	/	3	/	2.5
11	/	3.5	/	2.5	/	3
12	/	3	/	3	/	3
13	/	2.5	/	2	/	3
14	/	2.5	/	2	/	3
15	/	2	/	2	/	2.5
16	/	2.5	/	2.5	/	2.5
17	/	2.5	/	2.5	/	3
18	/	2	/	2	/	3
19	/	2.5	/	2.5	/	2.5
20	/	2.5	/	3	/	2.5
21	3.5	2.5	4	2.5	3.5	2.5
22	3	2.5	3	3	3.5	2.5
23	3.5	3.5	4	3.5	3	3
24	3.5	3.5	3.5	3.5	4	3.5
25	3.5	3	4.5	3.5	3	3
26	3.5	3	3.5	2.5	3	3.5
27	4.5	3.5	5	3.5	3.5	4
28	3.5	3	3.5	3	3.5	3
29	4	4	4	4.5	3	3
30	4	4	4	4.5	3.5	4.5
31	5	4	6	4	4.5	4
32	5	4	5	3.5	4.5	4
33	4	2.5	5.5	3	3	3.5
34	5	3	5.5	3.5	3.5	3
35	4	4	4.5	4.5	4.5	4
36	3.5	3.5	3.5	3.5	3	3
37	4.5	4	4.5	4.5	3.5	4
38	4.5	3.5	4.5	3.5	4.5	3.5
39	3	2.5	4	3	3.5	3
40	3.5	5	3.5	5	4	5
41	/	2.5	/	2	/	3
42	/	3	/	2	/	3
43	/	2.5	/	2.5	/	2.5
44	/	2.5	/	2.5	/	2.5
45	/	3	/	2.5	/	3
46	/	3	/	2.5	/	3.5
47	/	3	/	3	/	3.5
48	/	2.5	/	3	/	2.5
49	/	2.5	/	2.5	/	3
50	/	3.5	/	2.5	/	3
51	/	3	/	3	/	3
52	/	3.5	/	3	/	3.5
53	/	4	/	4	/	3.5
54	/	3.5	/	3.5	/	3.5
55	/	2.5	/	2.5	/	2.5
56	/	3	/	3	/	3.5
57	/	4	/	4.5	/	4.5
58	/	2.5	/	2.5	/	2
59	/	3	/	3	/	3
60	/	2.5	/	2.5	/	3
61	4.5	4	5.5	4	4.5	4
62	4.5	4	5	4.5	4.5	4
63	5	5	5	5	4.5	4.5
64	5	4.5	5.5	4.5	5	4.5
65	5.5	4.5	6	4	6	5
66	4.5	3	4.5	3	4.5	3
67	4.5	3.5	5.5	4	3.5	3
68	4	3	5	3	3.5	3
69	3.5	3	4	3	4	3.5
70	4	3	4.5	3	3.5	3.5
71	3.5	3.5	3.5	3.5	4	4
72	5.5	4.5	6	4.5	4	4
73	5	4	6	4	4.5	4
74	4.5	4	4.5	4.5	4	3.5
75	3.5	3.5	3.5	3	4	4.5
76	3	3	3.5	3	3.5	3
77	2.5	4	2.5	4	3	4
78	5	3.5	6	3.5	3.5	3.5
79	4.5	3.5	5.5	4	4	4
80	4	4	5	4	4.5	4

**Table 3 sensors-24-05341-t003:** Comparison results of various weight adjustment strengths.

Scene	δ = 2 True Positive	δ = 2 False Positive	δ = 3 True Positive	δ = 3 False Positive	δ = 4 True Positive	δ = 4 False Positive
1	/	2	/	1.5	/	2.5
2	/	2.5	/	2.5	/	3
3	/	2.5	/	2.5	/	2.5
4	/	3	/	3	/	3
5	/	2.5	/	2	/	2.5
6	/	2	/	2	/	2.5
7	/	2	/	2	/	2.5
8	/	2.5	/	2	/	2.5
9	/	2	/	2	/	2.5
10	/	3	/	3	/	2.5
11	/	3	/	2.5	/	3.5
12	/	3	/	3	/	3
13	/	2.5	/	2	/	3
14	/	2	/	2	/	2.5
15	/	2	/	2	/	2
16	/	2.5	/	2.5	/	2.5
17	/	2.5	/	2.5	/	3
18	/	2.5	/	2	/	2.5
19	/	2.5	/	2.5	/	2.5
20	/	3	/	3	/	3
21	3.5	2.5	4	2.5	4	3.5
22	3	3	3	3	3.5	3.5
23	3.5	3	4	3.5	3	3.5
24	3.5	3.5	3.5	3.5	4	3.5
25	3.5	3.5	4.5	3.5	3.5	3
26	3	2.5	3.5	2.5	3	3
27	4	3.5	5	3.5	4.5	4
28	3.5	3	3.5	3	3.5	4
29	4	3	4	4.5	3.5	5
30	4	4	4	4.5	4	4.5
31	4.5	3.5	6	4	5	4.5
32	4.5	3.5	5	3.5	4.5	4
33	4.5	3	5.5	3	5.5	4
34	5	3	5.5	3.5	4.5	4.5
35	4	4	4.5	4.5	4.5	4
36	3	3	3.5	3.5	4.5	4.5
37	4.5	4	4.5	4.5	4	4
38	4	3.5	4.5	3.5	5	4
39	3.5	3	4	3	3.5	3
40	3.5	5	3.5	5	4.5	5.5
41	/	2.5	/	2	/	3
42	/	2.5	/	2	/	3
43	/	2.5	/	2.5	/	3
44	/	2.5	/	2.5	/	3
45	/	2.5	/	2.5	/	3
46	/	2.5	/	2.5	/	3
47	/	3	/	3	/	3.5
48	/	2.5	/	3	/	2.5
49	/	2.5	/	2.5	/	3
50	/	3.5	/	2.5	/	3
51	/	3	/	3	/	3
52	/	3	/	3	/	4
53	/	3.5	/	4	/	3
54	/	3.5	/	3.5	/	3.5
55	/	2.5	/	2.5	/	2.5
56	/	3	/	3	/	4
57	/	4	/	4.5	/	4
58	/	2.5	/	2.5	/	2.5
59	/	3	/	3	/	3
60	/	2.5	/	2.5	/	3.5
61	4.5	4	5.5	4	4.5	4
62	4	4	5	4.5	4.5	4.5
63	4.5	4.5	5	5	4.5	4.5
64	5	4.5	5.5	4.5	5	4.5
65	5.5	4.5	6	4	5.5	4.5
66	4.5	3	4.5	3	4	3.5
67	4.5	3.5	5.5	4	4.5	4.5
68	4	3.5	5	3	4.5	3.5
69	3.5	3	4	3	4.5	3.5
70	4	3.5	4.5	3	4.5	3.5
71	3.5	3.5	3.5	3.5	4	4
72	5.5	4	6	4.5	4	4
73	5	4	6	4	5	3.5
74	4.5	4	4.5	4.5	4.5	4
75	3.5	3	3.5	3	4	4
76	3	3	3.5	3	3.5	3.5
77	3	3.5	2.5	4	3.5	5
78	5	3.5	6	3.5	5	4
79	4.5	4	5.5	4	4.5	4
80	4	4	5	4	4.5	4

**Table 4 sensors-24-05341-t004:** Comparison results of ablation experiments. OT, original true positive; OF, original false positive; PT, pyramid true positive; PF, pyramid false positive; WLT, weight learning true positive; WLF, weight learning false positive; CT, combined true positive; CF, combined true positive.

Scene	OT	OF	PT	PF	WLT	WLF	CT	CF
1	/	2.5	/	2.5	/	2	/	1.5
2	/	2.5	/	2.5	/	2.5	/	2.5
3	/	2.5	/	2.5	/	2.5	/	2.5
4	/	2.5	/	2.5	/	3	/	3
5	/	2.5	/	2.5	/	2	/	2
6	/	2.5	/	2.5	/	2.5	/	2
7	/	2	/	2.5	/	2.5	/	2
8	/	2.5	/	2.5	/	2.5	/	2
9	/	2	/	2	/	2.5	/	2
10	/	3	/	3	/	3	/	3
11	/	3	/	3	/	3.5	/	2.5
12	/	3	/	2.5	/	3.5	/	3
13	/	3	/	3	/	2.5	/	2
14	/	2.5	/	3	/	2.5	/	2
15	/	3	/	2.5	/	2.5	/	2
16	/	2.5	/	2.5	/	2.5	/	2.5
17	/	3	/	2.5	/	3	/	2.5
18	/	2.5	/	2.5	/	2.5	/	2
19	/	3	/	3	/	3	/	2.5
20	/	2.5	/	2.5	/	3	/	3
21	3	3	3	3	4	3.5	4	2.5
22	3	3	3	3	3	3	3	3
23	3	3	3	2.5	4	4	4	3.5
24	3.5	3.5	3	3	4	3.5	3.5	3.5
25	3.5	3.5	3.5	3	4	3.5	4.5	3.5
26	3	2.5	3	3	3.5	3	3.5	2.5
27	3.5	3.5	3.5	3	4.5	3.5	5	3.5
28	3.5	3	3.5	3	3.5	3.5	3.5	3
29	3	3	3	2.5	3	3	4	4.5
30	4	3.5	3.5	3.5	4	4	4	4.5
31	3	3	3	3	5	4	6	4
32	3.5	3.5	3.5	3.5	5	4.5	5	3.5
33	3.5	3	3.5	3	4	3.5	5.5	3
34	3	3	3	3	3.5	3.5	5.5	3.5
35	3.5	3.5	4	3.5	4.5	4	4.5	4.5
36	3	3	3.5	3	3.5	3.5	3.5	3.5
37	4	3.5	3.5	3.5	3.5	3.5	4.5	4.5
38	3.5	3.5	3.5	3.5	4	4	4.5	3.5
39	3.5	3	3	3	3.5	3.5	4	3
40	3.5	4.5	3.5	4.5	4	5	3.5	5
41	/	3	/	3	/	3	/	2
42	/	3	/	3	/	3	/	2
43	/	3	/	3	/	3	/	2.5
44	/	3	/	3	/	3	/	2.5
45	/	2.5	/	3	/	3	/	2.5
46	/	2.5	/	2.5	/	3	/	2.5
47	/	3	/	3	/	3	/	3
48	/	3	/	2.5	/	3	/	3
49	/	2.5	/	2.5	/	3	/	2.5
50	/	3	/	3	/	3	/	2.5
51	/	3.5	/	3	/	3.5	/	3
52	/	3	/	3	/	3.5	/	3
53	/	3.5	/	3.5	/	4	/	4
54	/	3.5	/	3.5	/	4	/	3.5
55	/	2.5	/	3	/	3	/	2.5
56	/	3	/	3	/	3.5	/	3
57	/	4	/	3.5	/	4	/	4.5
58	/	3	/	3	/	3	/	2.5
59	/	3.5	/	3.5	/	3.5	/	3
60	/	2.5	/	3	/	3.5	/	2.5
61	4	4	3.5	3.5	4.5	4	5.5	4
62	3.5	3	3.5	3.5	4.5	4.5	5	4.5
63	3.5	3.5	3.5	3	4	4	5	5
64	4.5	4	4	3.5	5	4.5	5.5	4.5
65	4	3.5	3.5	3	5	3.5	6	4
66	4	3	4	3	4	3	4.5	3
67	3.5	3.5	4	3.5	4.5	4	5.5	4
68	3.5	3.5	4	3.5	4.5	4	5	3
69	3	3	3	3	3.5	3.5	4	3
70	3.5	3.5	3.5	3	4	3.5	4.5	3
71	3.5	3	4	4	4	4	3.5	3.5
72	4	3.5	4.5	4	5	4	6	4.5
73	4.5	4	4	3	4.5	3.5	6	4
74	4	4	3.5	3.5	4.5	4	4.5	4.5
75	3.5	3	3.5	3	4	4	3.5	3
76	3	3	3	3	3.5	3	3.5	3
77	3	3.5	3.5	4	4	4.5	2.5	4
78	4	3.5	4	3	5	3.5	6	3.5
79	3.5	3.5	4	3	4	3	5.5	4
80	3.5	3.5	4	3.5	4.5	4	5	4

**Table 5 sensors-24-05341-t005:** Detection results of comparison experiments. LBPT, LBP method true positive; LBPF, LBP method false positive; MNMIT, MNMI method true positive; MNMIF, MNMI method false positive; HCT, Haralick coded method true positive; HCF, Haralick coded method false positive; PT, proposed method true positive; PF, proposed method false positive.

Scene	LBPT	LBPF	MNMIT	MNMIF	HCT	HCF	PT	PF
1	/	3	/	2.5	/	3	/	1.5
2	/	2	/	2.5	/	5.5	/	2.5
3	/	2.5	/	1.5	/	3	/	2.5
4	/	2	/	3	/	3	/	3
5	/	4	/	3	/	4	/	2
6	/	3.5	/	2.5	/	4.5	/	2
7	/	2	/	2	/	3.5	/	2
8	/	2	/	2	/	3.5	/	2
9	/	2.5	/	2.5	/	3	/	2
10	/	3.5	/	3.5	/	4	/	3
11	/	2.5	/	2	/	3.5	/	2.5
12	/	2	/	3.5	/	3.5	/	3
13	/	3.5	/	3	/	4.5	/	2
14	/	3	/	3.5	/	4.5	/	2
15	/	2	/	1.5	/	4	/	2
16	/	2	/	2	/	3.5	/	2.5
17	/	3	/	3.5	/	3.5	/	2.5
18	/	3.5	/	2	/	4	/	2
19	/	2	/	2.5	/	4	/	2.5
20	/	2.5	/	3	/	3.5	/	3
21	0	3	2	3	3.5	3.5	4	2.5
22	2	2	3	3	2	4	3	3
23	4	2	2.5	2	0	5.5	4	3.5
24	3	3	2.5	2.5	3	4	3.5	3.5
25	0	4	0.5	3	4.5	4	4.5	3.5
26	0	3	0.5	4	1.5	4.5	3.5	2.5
27	2.5	2.5	3.5	3	2.5	4	5	3.5
28	2.5	3	3.5	2	2.5	4	3.5	3
29	2	3.5	3	3	3	5	4	4.5
30	1.5	4.5	0	3.5	3.5	5	4	4.5
31	4	3	5	1.5	3.5	4	6	4
32	3.5	2.5	3.5	3.5	4	5	5	3.5
33	2.5	4	3.5	3.5	3.5	4.5	5.5	3
34	2	4.5	3.5	4	4	4	5.5	3.5
35	1.5	3.5	2	2.5	3.5	5	4.5	4.5
36	2	2	3	2.5	3.5	4	3.5	3.5
37	0	3	1	4.5	2.5	4	4.5	4.5
38	0	4	0.5	2.5	4	4	4.5	3.5
39	2	2.5	3	3	3	4	4	3
40	2.5	3	3	3.5	3.5	4	3.5	5
41	/	4	/	3	/	4	/	2
42	/	4	/	3.5	/	5.5	/	2
43	/	4.5	/	3	/	4.5	/	2.5
44	/	4	/	3	/	5.5	/	2.5
45	/	4	/	3	/	4.5	/	2.5
46	/	4	/	3	/	6	/	2.5
47	/	4	/	3	/	3	/	3
48	/	5	/	3.5	/	3.5	/	3
49	/	4	/	3	/	5	/	2.5
50	/	4.5	/	3	/	4	/	2.5
51	/	4	/	3	/	3.5	/	3
52	/	4.5	/	3.5	/	3.5	/	3
53	/	3	/	3	/	5	/	4
54	/	5	/	3	/	5	/	3.5
55	/	4	/	3	/	4	/	2.5
56	/	5	/	3	/	3.5	/	3
57	/	3.5	/	3	/	3	/	4.5
58	/	2.5	/	3	/	4	/	2.5
59	/	3	/	3	/	4	/	3
60	/	4.5	/	3.5	/	3.5	/	2.5
61	4.5	4.5	4.5	2.5	3	4	5.5	4
62	2	4	3	4	4.5	4	5	4.5
63	2.5	4.5	3	4	2	4.5	5	5
64	2	4	3	4	4	4	5.5	4.5
65	5	4.5	5	3.5	1.5	4.5	6	4
66	4	4	4	4	4.5	4.5	4.5	3
67	4	4	3.5	3.5	1.5	4	5.5	4
68	4	4	3.5	3.5	5.5	4.5	5	3
69	3.5	3.5	3	4	4	5	4	3
70	1.5	4	3	3.5	4	4	4.5	3
71	3.5	3.5	4	4	5.5	5	3.5	3.5
72	4	4	4	4	5.5	4.5	6	4.5
73	3.5	3.5	4.5	4.5	3	4	6	4
74	4.5	4.5	3.5	3.5	6	5	4.5	4.5
75	4	4	4	4	4	4	3.5	3
76	4	4	4.5	4.5	4	4	3.5	3
77	4	4	3.5	3.5	4	4	2.5	4
78	4	2	3.5	4.5	4.5	3.5	6	3.5
79	3.5	4	3.5	3.5	4.5	4.5	5.5	4
80	3.5	4	4	4	3.5	4.5	5	4

## Data Availability

Data are contained within the article.

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
