# Peer review of "An Anti-Forensics Video Forgery Detection Method Based on Noise Transfer Matrix Analysis"

_sensors, 2024, doi:10.3390/s24165341_

Round 1
Reviewer 1 Report
Comments and Suggestions for Authors
1、The meaning of some abbreviations is unclear, such as FRR interval?
2、Is n the number of frames in formulas 7 and 8? What is the operation for ln of formula 9?
3、The description of the subsection of dataset is inconsistent with the content of subsequent experiments
4、Lack of comparison with the following methods:
[31] Feng C, Xu Z, Jia S, et al. Motion-adaptive frame deletion detection for digital video forensics[J]. IEEE Transactions on Circuits and Systems for video Technology, 2016, 27(12): 2543-2554.
Reviewer 2 Report
Comments and Suggestions for Authors
In this paper, a video tampering detection approach is proposed. This method composes of three main steps: 1) extracting noise 2) computing the noise transfer matrix 3)adjusting the weights of the transfer matrix. This method is different from previous methods and demonstrates good empirical performances. However, there are several questions can be resolved to further improve the paper.
1. Clarify on the definition of GOP. The GOP term appears often in the paper. Maybe adding a figure in the beginning can explain the term GOP. And how GOP is determined from frames of videos.
2. Notation consistency: why \bar{I_{x,y,m} the average intensity in position (x,y), if it is the average intensity, it should be irrelevant to the position.
3. Why is Figure 3 the noise? It seems to be the outline of figures? Indeed, this method may measure the outline transfer instead of noise. Maybe another word than noise is better? Not very sure about it.
4. How the correspondence in Eq 2 is achieved?
5. Please consider discussing related fields ood/anomaly detection in your paper?
[1] Generalized Out-of-Distribution Detection: A Survey
[2] CRoFT: Robust Fine-Tuning with Concurrent Optimization for OOD Generalization and Open-Set OOD Detection
Overall, this is an interesting paper and the reviewer recommends minor revision to further improve the paper presentation.
Comments on the Quality of English LanguageThe English needs minor improvements
Round 2
Reviewer 1 Report
Comments and Suggestions for Authors
The authors have addressed my questions, So I suggest accepting it for publication.